

# Symmetries versus the spectrum of $J\bar{T}$ - deformed CFTs

**Monica Guica**[1,2,3]

**1** Institut de Physique Théorique, CEA Saclay, CNRS, 91191 Gif-sur-Yvette, France
**2** Department of Physics, Stockholm University, AlbaNova, 106 91 Stockholm, Sweden
**3** Nordita, Roslagstullsbacken 23, SE-106 91 Stockholm, Sweden

## Abstract

It has been recently shown that classical $J\bar{T}$ - deformed CFTs possess an infinite-dimensional Witt-Kac-Moody symmetry, generated by certain field-dependent coordinate and gauge transformations. On a cylinder, however, the equal spacing of the descendants' energies predicted by such a symmetry algebra is inconsistent with the known finite-size spectrum of $J\bar{T}$ - deformed CFTs. Also, the associated quantum symmetry generators do not have a proper action on the Hilbert space. In this article, we resolve this tension by finding a new set of (classical) conserved charges, whose action is consistent with semiclassical quantization, and which are related to the previous symmetry generators by a type of energy-dependent spectral flow. The previous inconsistency between the algebra and the spectrum is resolved because the energy operator does not belong to the spectrally flowed sector.

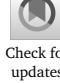
## Contents



# 1. Introduction and statement of the problem

The discovery of the AdS/CFT correspondence [1] has marked a major step in our current understanding of quantum gravity. While there are good reasons to believe that gravity in general backgrounds is holographic, various clues point towards the fact that for asymptotically flat spacetimes [2] or spacetimes related to the near-horizon geometry of extremal black holes [3], the dual QFT may be non-local. However, non-local quantum field theories are still relatively poorly understood, in comparison with their local counterparts.

In [4], Smirnov and Zamolodchikov (see also [5]) laid out the construction of a set of tractable irrelevant deformations of two-dimensional local QFTs which result into QFTs that are non-local, yet appear to be UV complete [6,7]. Moreover, these theories are solvable, in the sense that one can compute their spectrum, $S$-matrix and other observables [8–11] in terms of the corresponding quantities in the undeformed QFT. Even more interestingly, certain single-trace analogues of the Smirnov-Zamolodchikov deformations have been related to holography in non-asymptotically AdS spacetimes [12–14].

The Smirnov-Zamolodchikov deformations are constructed from bilinears of two conserved currents. The best studied of these is the so-called $T\bar{T}$ deformation, constructed from the components of the stress tensor. Deformations constructed from a $U(1)$ current and the stress tensor, such as the $J\bar{T}$ [15] and the $JT_a$ [16] deformations are also relatively well studied. Of these, the $J\bar{T}$ deformation of two-dimensional CFTs is the simplest, as the non-locality of the deformed QFT is concentrated exclusively to the right-moving side, and the theory stays local and conformal on the left. The effect of performing several of these deformations simultaneously has been studied in e.g. [17,18].

It is interesting to better understand the structure of the Smirnov-Zamolodchikov deformations from a quantum-field-theoretical point of view. It has been recently shown [19] that at least at the classical level, $T\bar{T}$, $J\bar{T}$ and $JT_a$ - deformed CFTs all posses an infinite-dimensional set of field-dependent symmetries, whose algebra consists of two commuting copies of the Witt or the Witt-Kac-Moody algebra, if $U(1)$ currents are present. This structure was suggested by the previous holographic analyses of [20] for $T\bar{T}$ and [21] for $J\bar{T}$. These analyses also allowed for the computation of the central extension of the symmetry algebra, which becomes Virasoro-Kac-Moody. If these symmetries survive quantization, then we would conclude that $T\bar{T}$, $J\bar{T}$ and $JT_a$ - deformed CFTs correspond to a non-local version of two-dimensional CFTs, with a similarly rigid structure that would highly deserve further exploration.

There is, however, a problem, which can be seen already at the semiclassical level. The symmetry analysis of [19] is valid on both the plane and the cylinder. In the latter case, one immediately encounters a tension between the equally-spaced energies of the Virasoro descendants predicted by the symmetry analysis and the energies of the deformed eigenstates in $T\bar{T}$, $J\bar{T}$ and $JT_a$ - deformed CFTs, which take a square root form. In this note, we address this issue for the simplest case of the $J\bar{T}$ deformation, where the locality of the left-moving side provides a useful guiding principle for finding its resolution.

To state the problem explicitly, we start with a review of the relevant facts. The finite-size energy spectrum of $J\bar{T}$ - deformed CFTs placed on a cylinder of circumference $R$ is given

by [14, 21]

$$E_R \equiv \frac{E-P}{2} = \frac{2}{\lambda^2}\left(R - \lambda J_0 - \sqrt{(R-\lambda J_0)^2 - \lambda^2 E_R^{(0)}R}\right), \quad E_L = E_R + P, \qquad (1.1)$$

where $\lambda$ is the deformation parameter (with dimensions of length), $J_0$ is the left-moving charge in the undeformed CFT, $P$ is the quantized momentum, $E_R^{(0)}$ is the undeformed right-moving energy and the chiral anomaly coefficient has been set to $2\pi$. Note that while in the undeformed CFT, the energies of the right-moving (Virasoro and Kac-Moody) descendats of a primary state are equally-spaced, those of the corresponding $J\bar{T}$-deformed descendants are not. The energies of the left-moving descendants do have equal spacing, since they are obtained by raising $E_L = E_R + P$ with $E_R^{(0)}$ and $J_0$ held fixed.

We note in passing that the relation between the undeformed and deformed energies can be suggestively written as spectral flow [22], with a parameter $\lambda E_R$ proportional to the right-moving energy

$$E_L R = E_L^{(0)}R + \lambda J_0 E_R + \frac{\lambda^2}{4}E_R^2, \qquad E_R(R - \lambda w) = E_R^{(0)}R + \lambda \bar{J}_0 E_R + \frac{\lambda^2}{4}E_R^2. \qquad (1.2)$$

Here, $\bar{J}_0$ is the right-moving $U(1)$ charge in the undeformed CFT, and $w = J_0 - \bar{J}_0$ is the winding charge. This observation will be quite useful later. It has already been used in deriving the spectrum in presence of a chiral anomaly [14] and for organising the conformal dimensions of $J\bar{T}$-deformed CFTs on the plane [23].

The symmetries of $J\bar{T}$ - deformed CFTs consist of, first, an infinite set of left-moving conformal and $U(1)$ gauge symmetries that enhance the $SL(2,\mathbb{R})_L \times U(1)_L$ global symmetries of the theory. These symmetries are parametrized by two arbitrary functions of the left-moving coordinate, $U = \sigma + t$. In the general Hamiltonian framework for $J\bar{T}$ - deformed CFTs developed in [19], they are generated by

$$Q_f = \int d\sigma f(U/R)\mathcal{H}_L, \qquad P_\eta = \int d\sigma \eta(U/R)(\mathcal{J}_+ + \frac{\lambda}{2}\mathcal{H}_R), \qquad (1.3)$$

where, in order for the argument of the functions to have periodicity one, the coordinate $U$ has been divided by the circumference of the circle. $\mathcal{H}_L = \mathcal{H}_R + \mathcal{P}$ is the left-moving Hamiltonian density, where the right-moving Hamiltonian density $\mathcal{H}_R$ is given in terms of its undeformed counterpart $\mathcal{H}_R^{(0)}$ by a formula entirely analogous to (1.1), with $J_0$ replaced by $\mathcal{J}_+$, the left-moving current density. The commutation relations of the deformed generators are then fixed by those of the undeformed currents, and one can show that the left-moving charge algebra is precisely Witt-Kac-Moody

$$\{Q_f, Q_g\} = \frac{1}{R}Q_{fg'-f'g}, \qquad \{Q_f, P_\eta\} = \frac{1}{R}P_{f\eta'}, \qquad \{P_\eta, P_\chi\} = \frac{1}{2}\int d\sigma \chi \partial_\sigma \eta. \qquad (1.4)$$

The factor of $R$ can be absorbed into a rescaling of the pseudoconformal charges $\bar{Q}_{\bar{f}}$.

The second set of infinite-dimensional symmetries of $J\bar{T}$ - deformed CFTs are field-dependent, and are generated by functions of the field-dependent coordinate

$$v = \sigma - t - \lambda\phi, \qquad (1.5)$$

where $\phi$ is related to the current $J$ via $J = \star d\phi$. The conserved pseudo-conformal and U(1) charges are given by

$$\bar{Q}_{\bar{f}} = \int d\sigma \bar{f}\left(\frac{v}{R_v}\right)\mathcal{H}_R, \qquad \bar{P}_{\bar{\eta}}^{KM} = \int d\sigma \bar{\eta}\left(\frac{v}{R_v}\right)(\mathcal{J}_- + \frac{\lambda}{2}\mathcal{H}_R), \qquad (1.6)$$

where $R_v = R - \lambda w$ is the field-dependent radius of the field-depedent coordinate $v$, and the particular combination $\bar{P}^{KM}$ of the right-moving currents is singled out by its simple commutation relations. Remarkably, these charges entirely commute with the left-moving ones, and the charge algebra is still a functional Witt-Kac-Moody algebra

$$\{\bar{Q}_{\bar{f}}, \bar{Q}_{\bar{g}}\} = \frac{1}{R_v}\bar{Q}_{\bar{f}'\bar{g}-\bar{f}\bar{g}'}, \quad \{\bar{Q}_{\bar{f}}, \bar{P}^{KM}_{\bar{\eta}}\} = -\frac{1}{R_v}\bar{P}^{KM}_{\bar{f}\bar{\eta}'}, \quad \{\bar{P}^{KM}_{\bar{\chi}}, \bar{P}^{KM}_{\bar{\eta}}\} = -\frac{1}{2}\int d\sigma\, \bar{\chi}\partial_\sigma\bar{\eta}. \quad (1.7)$$

The word 'functional' above refers to the fact that the factors of $R_v$ can be absorbed into a redefinition of the $'$ to mean 'derivative with respect to $v$', rather than to the full rescaled argument of the functions $\bar{f}, \bar{\eta}$, as used above. In any case, when labeling the charges in terms of the Fourier modes of $\bar{f}, \bar{\eta}$, etc., the field-dependent radius will explicitly appear in the algebra[1]. While this algebra is not exactly Witt-Kac-Moody, it still predicts, upon quantization, an equally-spaced spectrum of descendants, which is incompatible with the energy formula (1.1).

It is in fact not hard to notice already from their classical Poisson brackets, that the right-moving charges (1.6) will not have a proper action on the semiclassical phase space of the theory, where the charges associated to the global $U(1)$ symmetry and the momentum are quantized. Concretely, the problem appears to lie in the commutators of the right-moving generators with the $U(1)$ charges

$$\{\bar{Q}_{\bar{f}}, J_0\} = \frac{1}{R_v}\int_0^R d\sigma\, \bar{f}'\left(\frac{v}{R_v}\right)\mathcal{H}_R(\sigma)\{v(\sigma), J_0\} = -\frac{\lambda}{2R_v}\bar{Q}_{\bar{f}'} = \{\bar{Q}_{\bar{f}}, \bar{J}_0\}, \quad (1.8)$$

and the analogous commutators of the right-moving $U(1)$ generators $\bar{P}_{\bar{\eta}}$ with $J_0$ and $\bar{J}_0$. What this means is that $J_0 + \bar{J}_0$, which represents the global $U(1)$ charge of the configuration and which should be quantized, is changed by a non-integer amount (more precisely, $2\pi\lambda n/R_v$, with $n \in \mathbb{Z}$) by the action of the semiclassically quantized right-moving generators on a state in the deformed theory. A similar statement holds for the momentum, which from (1.7) can be shown to satisfy

$$\{\bar{Q}_{\bar{f}}, P\} = -\frac{1}{R_v}\bar{Q}_{\bar{f}'}, \quad \{\bar{P}_{\bar{\eta}}, P\} = -\frac{1}{R_v}\bar{P}_{\bar{\eta}'}, \quad (1.9)$$

i.e. it is changed by units of $2\pi/R_v$, instead of $2\pi/R$. These observations imply that the action of the right-moving generators (with the exception of the global right-moving energy and charge) on a field configuration is in tension with semiclassical quantization. Hence, the naive quantum versions of the charges (1.6) do not act properly on the Hilbert space of $J\bar{T}$-deformed CFTs on a cylinder.

While having an infinite set of symmetry generators that do not properly act on the Hilbert space of the system is not very useful, an interesting question is whether these generators can be modified in such a way that their algebra is preserved, but their action on the Hilbert space is rectified. In this note, we show that this is indeed possible, by explicitly constructing an infinite set of charges that, upon quantization, would act on the deformed finite-size Hilbert space in a way consistent with charge and momentum quantization. These charges can therefore be used to organise the spectrum of the deformed CFT.

To find them, we study the flow equation with respect to the deformation parameter $\lambda$ of the various energy eigenstates and compare it to the flow of the symmetry generators (1.3) and (1.6). Introducing a new set of operators that relate deformed descendant states to the deformed primaries, we find that they are related to the previously discussed symmetry generators by a type of energy-dependent spectral flow transformation. The new symmetry generators are conserved and satisfy a Witt-Kac-Moody algebra with a field-independent radius.

---

[1] In the $J\bar{T}$ case, we can simply rescale the generators by $R_v$ to obtain a usual Witt-Kac-Moody algebra.

Their commutation relations with the energy and momentum are non-trivial though, as the latter two operators belong to the unflowed sector. This resolves the apparent tension between the symmetry algebra and the spectrum of $J\bar{T}$ - deformed CFTs. We should note that while our analysis is mostly classical - i.e., at the level of Poisson brackets - the quantum generalization of these generators now appears to be straightforward.

This paper is organised as follows. In section 2, we derive the flow equation satisfied by the energy eigenstates in a $J\bar{T}$ - deformed CFT, by adapting the method used in [11] to study the flow of states under the $T\bar{T}$ deformation. We subsequently compare this to the flow equations satisfied by the symmetry generators, and argue that the two sets of generators must be related by a similarity transformation that we denote as "spectral flow". In section 3, we proceed to finding the flowed operators, first perturbatively and then by making an all-orders proposal, whose consistency we then check. The technical details of the very many Poisson brackets we need are collected in the appendices.

## 2. Flow of the eigenstates versus the symmetry generators

### 2.1. The flow of energy eigenstates

Let $|n_\lambda\rangle$ be an energy (and momentum, and charge) eigenstate in the theory deformed by an amount $\lambda$. As $\lambda$ is infinitesimally changed, the change in the eigenstate is given by first-order quantum-mechanical perturbation theory

$$\partial_\lambda |n_\lambda\rangle = \sum_{m\neq n} \frac{\langle m_\lambda | \partial_\lambda H | n_\lambda \rangle}{E_n^\lambda - E_m^\lambda} |m_\lambda\rangle, \tag{2.1}$$

where $\partial_\lambda H$ is the change in the Hamiltonian. For convenience, we take the deforming operator to be $\tilde{J}\bar{T}$, rather than $J\bar{T}$, where $\tilde{J} = \star d\phi$ is a topologically conserved current. Its components are

$$\tilde{J}_t = \phi', \qquad \tilde{J}_\sigma = \partial_\pi \mathcal{H}, \tag{2.2}$$

where $\pi$ is the canonical momentum conjugate to $\phi$. One can easily check, using the method developed in [19], that the $\tilde{J}\bar{T}$ deformation leads to the same deformed Hamiltonian density as $J\bar{T}$. Consequently, the change in the Hamiltonian is given by[2]

$$\partial_\lambda H(\lambda) = -\int d\sigma \mathcal{O}_{\tilde{J}\bar{T}} = -\int d\sigma\, \epsilon^{\alpha\beta} \tilde{J}_\alpha(\sigma) T_{\beta V}(\sigma). \tag{2.3}$$

To find the general solution for the deformed eigenstates, we will use the technique proposed by [11]. On an equal-time slice, we write

$$\int d\sigma \mathcal{O}_{\tilde{J}\bar{T}} = \int d\sigma d\tilde{\sigma}\, \epsilon^{\alpha\beta} \tilde{J}_\alpha(\sigma) \delta(\sigma - \tilde{\sigma}) T_{\beta V}(\tilde{\sigma}). \tag{2.4}$$

It is useful to introduce the Green's function on the cylinder of circumference $R$

$$G(\sigma) = \frac{1}{2\pi i} \sum_{m\neq 0} \frac{1}{m} e^{2\pi i m\sigma/R} = \frac{1}{2} \mathrm{sgn}(\sigma) - \frac{\sigma}{R}, \tag{2.5}$$

which is single-valued and satisfies

$$\partial_\sigma G(\sigma - \tilde{\sigma}) = \delta(\sigma - \tilde{\sigma}) - \frac{1}{R}. \tag{2.6}$$

---

[2] We are using the conventions of [19] throughout this article.

Then, we can rewrite the deforming operator as

$$
\begin{aligned}
\int d\sigma \mathcal{O}_{\tilde{J}\bar{T}} &= \int d\sigma d\tilde{\sigma} \left[ \tilde{J}_\sigma(\sigma)\left(\frac{1}{R} + \partial_\sigma G(\sigma - \tilde{\sigma})\right) T_{tV}(\tilde{\sigma}) - \tilde{J}_t(\sigma)\left(\frac{1}{R} - \partial_{\tilde{\sigma}} G(\sigma - \tilde{\sigma})\right) T_{\sigma V}(\tilde{\sigma}) \right] \\
&= \frac{1}{R}\epsilon^{\alpha\beta}\tilde{J}_\alpha^0 T_{\beta V}^0 - \int d\sigma d\tilde{\sigma} G(\sigma - \tilde{\sigma})\left(\partial_\sigma \tilde{J}_\sigma(\sigma) T_{tV}(\tilde{\sigma}) + \tilde{J}_t(\sigma)\partial_{\tilde{\sigma}} T_{\sigma V}(\tilde{\sigma})\right) \\
&= \frac{1}{R}\epsilon^{\alpha\beta}\tilde{J}_\alpha^0 T_{\beta V}^0 - \partial_t \int d\sigma d\tilde{\sigma} G(\sigma - \tilde{\sigma})\tilde{J}_t(\sigma) T_{tV}(\tilde{\sigma}),
\end{aligned} \tag{2.7}
$$

where we introduced the notation

$$
\tilde{J}_\alpha^0 \equiv \int d\sigma \, \tilde{J}_\alpha \,, \qquad T_{\beta V}^0 \equiv \int d\sigma \, T_{\beta V}. \tag{2.8}
$$

Naturally, the integral of the time components of the currents above will yield the associated conserved charges, i.e. the winding, $w$, and respectively (minus) the right-moving energy, $-E_R$. We further use the manipulations of [11] to rewrite the regulated denominator of (2.1) as an integral, in terms of which the flow equation for the states becomes

$$
\begin{aligned}
\partial_\lambda |n_\lambda\rangle &= -i \sum_{m \neq n} \int_{-\infty}^0 dt \, e^{t\epsilon} |m_\lambda\rangle\langle m_\lambda|\partial_\lambda H_\lambda(t)|n_\lambda\rangle \\
&= i \sum_{m \neq n} |m_\lambda\rangle\langle m_\lambda| \frac{1}{R}\int_{-\infty}^0 dt \, e^{t\epsilon} \epsilon^{\alpha\beta}\tilde{J}_\alpha^0 T_{\beta V}^0 - \int d\sigma d\tilde{\sigma} G(\sigma - \tilde{\sigma})\tilde{J}_t(\sigma) T_{tV}(\tilde{\sigma}) \Big| n_\lambda\rangle .
\end{aligned} \tag{2.9}
$$

Here, $\epsilon > 0$ is an infinitesimal regulator used to make the integral converge, and the second term is evaluated on the $t = 0$ slice. Since we are working on the cylinder, the first term cannot be ignored. To evaluate it, we need the explicit form of the spatial components of $\tilde{J}$ and the right-moving translation generator, which can be worked out using the formulae given in [19]

$$
\tilde{J}_\sigma = \partial_\pi \mathcal{H} = \phi' + 2\partial_\pi \mathcal{H}_R = \phi' + 2\frac{\mathcal{J}_- + \lambda \mathcal{H}_R/2}{\sqrt{\phantom{x}}} \tag{2.10}
$$

and

$$
T_{\sigma V} = 2T_{VV} - \mathcal{H}_R = 2\frac{\mathcal{H}_R}{\sqrt{\phantom{x}}} - \mathcal{H}_R, \tag{2.11}
$$

where the somewhat unusual notation $\sqrt{\phantom{x}}$ is a shorthand for $\sqrt{(1 - \lambda\mathcal{J}_+)^2 - \lambda^2 \mathcal{H}_R^{(0)}}$.

Therefore,

$$
\epsilon^{\alpha\beta}\tilde{J}_\alpha^0 T_{\beta V}^0 = -2w \int d\sigma \frac{\mathcal{H}_R}{\sqrt{\phantom{x}}} - 2E_R \int d\sigma \frac{\mathcal{J}_- + \lambda\mathcal{H}_R/2}{\sqrt{\phantom{x}}}. \tag{2.12}
$$

In order to perform the time integral in (2.9), we would like to rewrite the above operator as a time derivative, i.e. as a commutator with the Hamiltonian. This can be achieved by introducing the zero modes

$$
\phi_0 \equiv \int_0^R d\sigma \, \phi(\sigma) \,, \qquad \chi_0 \equiv \int_0^R d\sigma \, \chi(\sigma), \tag{2.13}
$$

where the auxiliary non-local field $\chi$ is defined via

$$
\partial_\sigma \chi \equiv \mathcal{H}_R. \tag{2.14}
$$

Such fields also made their appearance in the analysis of the charge algebra for $T\bar{T}$ - deformed CFTs in [19], though they were not given an explicit name.

The Poisson brackets of the fields $\phi$ and $\chi$ (and, consequently, of their zero modes $\phi_0$ and $\chi_0$) are fixed by the Poisson brackets of the corresponding currents $\mathcal{J}_\pm, \mathcal{H}_R, \mathcal{P}$, up to some possible integration functions. While the choice of these functions is straightforward for the $\phi$ commutators, as $\phi$ is a local field, it is however somewhat subtle for the case of the $\chi$ commutators, because $\chi$ is non-local. In appendix B, we perform a rather detailed analysis of the Jacobi identities that constrain these integration functions, with the result

$$\{\chi_0, H\} = -2\int_0^R d\sigma \frac{\mathcal{H}_R}{\sqrt{}} + E_R + \frac{E_R R}{R_v}, \qquad \{\phi_0, H\} = 2\int_0^R d\sigma \frac{\mathcal{J}_- + \lambda/2\mathcal{H}_R}{\sqrt{}} + w, \qquad (2.15)$$

where $E_R = \int d\sigma \mathcal{H}_R$ is the right-moving energy operator. Consequently, we can write

$$\epsilon^{\alpha\beta} \tilde{J}_\alpha^0 T_{\beta V}^0 = -\{H, w\chi_0 - E_R\phi_0\} - wE_R\frac{R}{R_v} = \frac{d}{dt}(w\chi_0 - E_R\phi_0) - w\frac{E_R R}{R_v}. \qquad (2.16)$$

Plugging this into (2.9), the last term drops out, as it is evaluated between two different energy-momentum eigenstates. The integral of the first term over the half line yields, in the limit $\epsilon \to 0$

$$\partial_\lambda |n_\lambda\rangle = i\sum_{m \neq n} |m_\lambda\rangle\langle m_\lambda| \left. \frac{w\chi_0 - E_R\phi_0}{R} + \int d\sigma d\tilde{\sigma} G(\sigma - \tilde{\sigma})\phi'(\sigma)\mathcal{H}_R(\tilde{\sigma}) \right| n_\lambda\rangle. \qquad (2.17)$$

We will be denoting these two contributions as $\Delta\mathcal{O}$ and respectively $\hat{\mathcal{O}}$, defined as

$$\Delta\mathcal{O} \equiv \frac{w\chi_0 - E_R\phi_0}{R}, \qquad \hat{\mathcal{O}} \equiv \int d\sigma d\tilde{\sigma} G(\sigma - \tilde{\sigma})\phi'(\sigma)\mathcal{H}_R(\tilde{\sigma}), \qquad (2.18)$$

and their sum will be denoted as $\mathcal{O}_{tot} = \Delta\mathcal{O} + \hat{\mathcal{O}}$. If we make use of the identity

$$\int d\tilde{\sigma} \phi'(\tilde{\sigma}) G(\tilde{\sigma} - \sigma) = \phi(\tilde{\sigma}) G(\tilde{\sigma} - \sigma)|_0^R - \phi(\sigma) + \phi_0 = -\hat{\phi}(\sigma) + \phi_0, \qquad (2.19)$$

where $\hat{\phi}(\sigma) = \phi(\sigma) - w\sigma/R$ is the scalar field with its winding mode removed (which is thus single-valued on the circle), then an alternate expression for $\mathcal{O}_{tot}$ is

$$\mathcal{O}_{tot} = \frac{w\chi_0}{R} - \int d\sigma \hat{\phi}(\sigma)\mathcal{H}_R, \qquad (2.20)$$

which is rather useful in computing its Poisson brackets.

As a final step of our manipulations, we use the assumed completeness of the set of states to rewrite the flow equation for the energy eigenstates as

$$\partial_\lambda |n_\lambda\rangle = i\mathcal{O}_{tot}|n_\lambda\rangle - i|n_\lambda\rangle\langle n_\lambda|\mathcal{O}_{tot}|n_\lambda\rangle. \qquad (2.21)$$

Introducing an operator, $D$, which is diagonal in the energy eigenbasis and whose matrix elements are defined as $\langle n_\lambda|D|n_\lambda\rangle = \langle n_\lambda|\mathcal{O}_{tot}|n_\lambda\rangle$, we can rewrite the flow equation for the eigenstates in its final form

$$\boxed{\partial_\lambda |n_\lambda\rangle = i(\mathcal{O}_{tot} - D)|n_\lambda\rangle}. \qquad (2.22)$$

Thus, to understand the flow of the states, we need to understand also which parts of $\mathcal{O}_{tot}$ have non-zero expectation values in the energy eigenstates. This is a quite non-trivial task for arbitrary values of the flow parameter. We can nevertheless attempt to understand this problem perturbatively. For example, at $\lambda = 0$, we can use (2.5) to evaluate

$$i\hat{\mathcal{O}} = \sum_{m \neq 0} \frac{1}{m} : \tilde{J}_m \bar{L}_m : + \dots, \qquad (2.23)$$

where $\tilde{J}_m = J_m - \bar{J}_{-m}$ and $\bar{L}_m$ are the Fourier modes of $\phi'$ and respectively $\mathcal{H}_R$ in the undeformed CFT, and the colons denote normal ordering. Since the sum is strictly over non-zero modes, it is clear that the expectation value of this operator in any energy eigenstate of the undeformed CFT is zero. Thus, $\hat{\mathcal{O}}$ does not contribute to $D$, at least at $\lambda = 0$. On the other hand, the expectation value of $\Delta\mathcal{O}$ vanishes between any two different energy eigenstates at $\lambda = 0$, as one can see by evaluating

$$\langle m | [H, \Delta\mathcal{O}] | n \rangle = (E_m - E_n)\langle m | \Delta\mathcal{O} | n \rangle = i\langle m | (J_0 + \bar{J}_0)E_R | n \rangle = 0, \tag{2.24}$$

which implies that[3] $\langle m | \Delta\mathcal{O} | n \rangle = 0$, $\forall m \neq n$. Thus, we find that at $\lambda = 0$, $D = \Delta\mathcal{O}$.

At higher orders in perturbation theory, $\Delta\mathcal{O}$ may start having non-zero matrix elements between different eigenstates, which would therefore not contribute to $D$. To understand what happens, we should study the change with $\lambda$ of the matrix elements $\langle m | \Delta\mathcal{O} | n \rangle$

$$\partial_\lambda \langle m | \Delta\mathcal{O} | n \rangle = \langle m | \partial_\lambda \Delta\mathcal{O} - i[\mathcal{O}_{tot} - D, \Delta\mathcal{O}] | n \rangle = \langle m | \hat{\mathcal{D}}_\lambda \Delta\mathcal{O} | n \rangle + i(D_m - D_n)\langle m | \Delta\mathcal{O} | n \rangle, \tag{2.25}$$

where the flow operator $\hat{\mathcal{D}}_\lambda$ is defined as

$$\hat{\mathcal{D}}_\lambda \equiv \partial_\lambda - i[\mathcal{O}_{tot}, \cdot]. \tag{2.26}$$

Using the explicit expression, (3.3), for $\mathcal{D}_\lambda \Delta\mathcal{O}$ computed in the next section, we see that at $\lambda = 0$, the only contribution to $\langle n | \mathcal{D}_\lambda \Delta\mathcal{O} | n \rangle$ comes from the terms proportional to the zero modes of the fields $\phi$ and $\chi$. The $\lambda$ dependence of the diagonal matrix elements of $\hat{\mathcal{O}}$ can be studied by plugging in the known expression for $\mathcal{H}_R(\lambda)$. At first order in $\lambda$, it also does not look like this operator has non-zero diagonal matrix elements in the energy eigenbasis[4], and thus it will not contribute to $D$.

To summarize, up to first order in $\lambda$, we expect that

$$D = \Delta\mathcal{O} - \lambda(\mathcal{D}_\lambda \Delta\mathcal{O})_{no\,z.m.} + \mathcal{O}(\lambda^2), \tag{2.27}$$

i.e. we are subtracting all the off-diagonal contributions to $\Delta\mathcal{O}$ up to this order. Performing this analysis to higher order looks increasingly cumbersome, and we may need a better method.

The discussion so far holds for states defined on the $t = 0$ slice. It is interesting to also consider the flow equation for states defined at a time $t$ instead of $t = 0$. Our derivation of the flow operator (2.17) still holds, except that it should now be evaluated at time $t$, rather than $t = 0$. Since the states at $t$ are related to the states at $t = 0$ by a $\lambda$-dependent energy factor, the flow equation is best written as

$$\partial_\lambda |n_\lambda(t)\rangle = i(\mathcal{O}_{tot}(t) - \partial_\lambda E_n t - D(t))|n_\lambda(t)\rangle, \quad \partial_\lambda E = 2\frac{E_R Q_K}{R - \lambda Q_K}, \tag{2.28}$$

where $Q_K \equiv J_0 + \lambda E_R/2$, and the expression for $\partial_\lambda E$ is obtained from (1.1). The matrix elements of the operator $D(t)$ are defined as the expectation values of $\mathcal{O}_{tot}(t)$ in energy eigenstates.

## 2.2. Flow of the symmetry generators

Having understood the flow of the energy eigenstates with respect to $\lambda$, we would now like to discuss the corresponding flow of the symmetry generators $Q_f, P_\eta, \bar{Q}_{\bar{f}}$ and $\bar{P}_{\bar{\eta}}$. It is useful to compute the action of the operator $\mathcal{D}_\lambda$ defined above on these generators. Our analysis will

---

[3]For degenerate eigenstates, one can repeat the argument for the commutator with other globally conserved charges.

[4]Using the formulae in appendix C, it can be shown that $\hat{\mathcal{O}}$ satisfies the very simple flow equation $\mathcal{D}_\lambda \hat{\mathcal{O}} = w\hat{\mathcal{O}}/R_v$ (for $a = -1$), which can be used to evaluate its contribution, if any, at higher orders in $\lambda$.

be classical, and thus we will be using the Poisson bracket counterpart of this flow operator, i.e

$$\mathcal{D}_\lambda \equiv \partial_\lambda + \{\mathcal{O}_{tot}, \cdot\}, \tag{2.29}$$

obtained by the usual replacement $[\,,\,] \to i\{\,,\,\}$. To compute the action of $\mathcal{D}_\lambda$, we will need the Poisson brackets of the various currents in the $J\bar{T}$-deformed CFT, which were derived in [19] and are collected for convenience in appendix A. We will also need the Poisson brackets of the various symmetry currents with the zero modes of $\chi$ and $\phi$.

The commutators of the zero mode of $\phi$ are obtained by simply integrating the corresponding commutators of the field $\phi(\sigma)$, and we obtain

$$\{\phi_0, \mathcal{H}_R(\sigma)\} = \frac{\mathcal{J}_- + \frac{\lambda}{2}\mathcal{H}_R}{\sqrt{(1 - \lambda\mathcal{J}_+)^2 - \lambda^2\mathcal{H}_R^{(0)}}}\,, \qquad \{\phi_0, \mathcal{P}(\sigma)\} = \phi'(\sigma)\,, \qquad \{\phi_0, \mathcal{J}_\pm\} = \frac{1}{2}. \tag{2.30}$$

Note that in the CFT limit, $\{\phi_0, \mathcal{H}_L\} = \mathcal{J}_+$ and $\{\phi_0, \mathcal{H}_R\} = \mathcal{J}_-$, so the exponential of this operator is precisely what generates spectral flow for the left- and the right-movers.

The Poisson brackets of the zero mode $\chi_0$ are significantly more involved, due to the fact that the ancillary field $\chi$ is non-local, being defined as the integral of the local current $\mathcal{H}_R$. Consequently, its Poisson brackets are defined only up to certain integration functions, whose form is non-trivially constrained by various Jacobi identities. These constraints are analysed in detail in appendix B, and the end result for the various commutators of $\chi_0$ is

$$\{\chi_0, \mathcal{J}_+\} \;=\; -\frac{\lambda R}{2}\partial_\sigma\left[\frac{\mathcal{H}_R}{\sqrt{}}\left(1 + a - \frac{\lambda\hat{\phi}}{R_v}\right)\right] - \frac{\lambda R}{2R_v}\mathcal{H}_R, \tag{2.31}$$

$$\{\chi_0, \mathcal{J}_-\} \;=\; R\,\partial_\sigma\left[\frac{\mathcal{J}_-}{\sqrt{}}\left(1 + a - \frac{\lambda\hat{\phi}}{R_v}\right)\right] - \frac{\lambda R}{2R_v}\mathcal{H}_R, \tag{2.32}$$

$$\{\chi_0, \mathcal{H}_R\} \;=\; -\frac{\mathcal{H}_R}{\sqrt{}} + \frac{R}{R_v}\mathcal{H}_R + R\partial_\sigma\left(\frac{\mathcal{H}_R}{\sqrt{}}\left(1 + a - \frac{\lambda\hat{\phi}}{R_v}\right)\right), \tag{2.33}$$

$$\{\chi_0, \mathcal{P}\} \;=\; \mathcal{H}_R - \frac{R}{R_v}\mathcal{H}_R - R\partial_\sigma\left(\frac{\mathcal{H}_R}{\sqrt{}}\left(1 + a - \frac{\lambda\hat{\phi}}{R_v}\right)\right), \tag{2.34}$$

$$\{\chi_0, \phi\} \;=\; -\frac{\mathcal{J}_- + \lambda\mathcal{H}_R/2}{\sqrt{}}R\left(1 + a - \frac{\lambda\hat{\phi}}{R_v}\right), \tag{2.35}$$

where, as before, $\hat{\phi} = \phi - w\sigma/R$ equals $\phi$ with its winding mode removed. The terms proportional to the constant $a$ are allowed by all the Jacobi identities we have studied[5]. Since its value does not seem to be fixed and, moreover, it drops out from most of our subsequent computations, we will henceforth fix it to the convenient value $a = -1$.

Using these, one can compute the flow equations for the various currents, which are spelled out for convenience in appendix C, and from them derive the flow of the conserved charges. One finds that the left-moving charges are simply constant with respect to $\mathcal{D}_\lambda$

$$\mathcal{D}_\lambda Q_f = \mathcal{D}_\lambda P_\eta = 0, \tag{2.36}$$

while the right-moving ones satisfy

$$\mathcal{D}_\lambda \bar{Q}_{\bar{f}} = \frac{w}{R_v}\bar{Q}_{\bar{f}} - \frac{wt}{R_v^2}\bar{Q}_{\bar{f}'}\,, \qquad \mathcal{D}_\lambda \bar{P}_{\bar{\eta}}^{KM} = -\frac{wt}{R_v^2}\bar{P}_{\bar{\eta}'}^{KM}. \tag{2.37}$$

---

[5]This does not mean that there cannot exist other Jacobi identities that constrain the value of $a$, or that require the introduction of new terms in the commutators above. Our analysis is thus valid up to this caveat.

Note that the first term on the right-hand side of the $\bar{Q}_{\bar{f}}$ flow is necessary in order for the flow equation to be compatible with the charge algebra (1.7), as the latter contains an explicit factor of $1/R_v$, whose $\lambda$ derivative does not vanish. If we consider instead the rescaled dimensionless charges $R_v \bar{Q}_{\bar{f}}$, they satisfy a flow equation analogous to that of $\bar{P}^{KM}$. Their algebra is also the standard Witt-Kac-Moody algebra.

The explicit time dependence appearing on the right-hand side of (2.37) can be understood by computing the time derivative of e.g. $\bar{P}^{KM}$, where as usual $\frac{d}{dt} = \partial_t - \{H, \cdot\}$. One finds that $\frac{d}{dt} \bar{P}^{KM} = \{\mathcal{D}_\lambda H, \bar{P}_{\bar{\eta}}\} \neq 0$ because, as a result of the first equation, $\mathcal{D}_\lambda H = \omega E_R / R_v$.

Given the above form of the flow equations, it is convenient to define

$$\mathcal{D}'_\lambda = \mathcal{D}_\lambda - \frac{w E_R t}{R_v}, \tag{2.38}$$

which annihilates all of the (rescaled) conserved charges.

## 2.3. Relating the two

To summarize, we found that the quantum version of the rescaled conserved charges $Q_f, P_\eta$, $R_v \bar{Q}_{\bar{f}}$ and $\bar{P}^{KM}_{\bar{\eta}}$, which we will collectively denote as $\mathcal{L}$, are annihilated by the operator (2.29)

$$\hat{\mathcal{D}}_\lambda \mathcal{L} = \partial_\lambda \mathcal{L} - i[\mathcal{O}_{tot}, \mathcal{L}] = 0, \tag{2.39}$$

(or $\hat{\mathcal{D}}'_\lambda \mathcal{L} = 0$ if we work at $t \neq 0$). On the other hand, the states satisfy the flow equation (2.22), which involves an additional diagonal operator $D$, which can be rather complicated.

We now consider two energy eigenstates, $|n_0\rangle$ and $|n'_0\rangle$, that in the undeformed CFT are related by the action of a symmetry generator, $|n'_0\rangle = \mathcal{L}_{(\lambda=0)}|n_0\rangle$, which can be any of the Virasoro or Kac-Moody generators. Our goal is to find a new operator, $\widetilde{\mathcal{L}}$, that relates the corresponding flowed states in the deformed CFT, i.e. $|n'_\lambda\rangle = \widetilde{\mathcal{L}}|n_\lambda\rangle$. The flow equation (2.22) for the states then implies that the flow equation for the corresponding operators is

$$\partial_\lambda \widetilde{\mathcal{L}} - i[\mathcal{O}_{tot} - D, \widetilde{\mathcal{L}}] = 0. \tag{2.40}$$

The solutions to the two flow equations are related by $\widetilde{\mathcal{L}} = e^X \mathcal{L} e^{-X}$ where $X$ must satisfy

$$[\widetilde{\mathcal{L}}, (\partial_\lambda e^X - i[\mathcal{O}_{tot}, e^X])e^{-X} - D] = 0 \tag{2.41}$$

for *any* $\widetilde{\mathcal{L}}$. This implies that the second argument either vanishes, or it is proportional to the identity or some other operator that commutes with all the $\widetilde{\mathcal{L}}$. Assuming for simplicity that it vanishes, we can write

$$D = (\partial_\lambda e^X - i[\mathcal{O}_{tot}, e^X]) e^{-X}. \tag{2.42}$$

Noting that

$$(\partial_\lambda e^X) e^{-X} = \partial_\lambda X + \frac{1}{2}[X, \partial_\lambda X] + \frac{1}{3!}[X, [X, \partial_\lambda X]] + \dots \tag{2.43}$$

$$e^X \mathcal{O}_{tot} e^{-X} = \mathcal{O}_{tot} + [X, \mathcal{O}_{tot}] + \frac{1}{2}[X, [X, \mathcal{O}_{tot}]] + \dots, \tag{2.44}$$

the above equation can be written as

$$D = \hat{\mathcal{D}}_\lambda X + \frac{1}{2}[X, \hat{\mathcal{D}}_\lambda X] + \frac{1}{3!}[X, [X, \hat{\mathcal{D}}_\lambda X]] + \dots, \qquad \hat{\mathcal{D}}_\lambda X \equiv \partial_\lambda X - i[\mathcal{O}_{tot}, X]. \tag{2.45}$$

This result gives us a way to construct $X$, and therefore $\widetilde{\mathcal{L}}$, if we know $\mathcal{L}$ and $D$. If we work at $t \neq 0$, then $\mathcal{D}_\lambda$ should be replaced by $\mathcal{D}'_\lambda$, $\mathcal{O}_{tot}$ by $\mathcal{O}_{tot}(t) - w E_R t / R_v$ and $D$ by $D_{tot}(t) \equiv D(t) + \partial_\lambda E t$, as follows from (2.28).

As we already explained, finding $D$ to all orders is a rather difficult task, but we can certainly attempt this exercise perturbatively. Since at $\lambda = 0$, $D = \Delta\mathcal{O} = (w\chi_0 - E_R\phi_0)/R$, where $\phi_0$ is known to implement spectral flow in a CFT, we will henceforth denote the $\widetilde{\mathcal{L}}$ as the "spectrally flowed" operators, in this case by an energy-dependent amount. This connection will be made significantly more precise in the next section.

## 3. The spectrally flowed generators

### 3.1. Perturbative construction of the spectrally flowed generators

In this section, we attempt to solve the equation (2.45) for $X$ perturbatively, for $D$ given in (2.27), and use the solution to find the first few terms in the $\lambda$ expansion of the flowed generators. This can be done by assuming $X$ has an expansion of the form

$$X = \lambda\mathcal{O}_1 + \lambda^2\mathcal{O}_2 + \ldots \quad \Rightarrow \quad \mathcal{D}_\lambda X = \mathcal{O}_1 + \lambda\mathcal{D}_\lambda\mathcal{O}_1 + 2\lambda\mathcal{O}_2 + \lambda^2\mathcal{D}_\lambda\mathcal{O}_2 + \ldots, \tag{3.1}$$

where the $\mathcal{O}_n$ are in general non-linear functions of $\lambda$. We would moreover like to work at $t \neq 0$ so, according to our previous discussion, $\mathcal{D}_\lambda$ should be replaced by $\mathcal{D}'_\lambda$ and

$$D_{tot}(t) \equiv D(t) + \partial_\lambda E\, t = D_0 - \lambda(\mathcal{D}'_\lambda D_0)_{no\, z.m.} + \mathcal{O}(\lambda^2) + \frac{2Q_K E_R}{R - \lambda Q_K}\,t, \tag{3.2}$$

where $D_0 = \Delta\mathcal{O} - wE_R t/R_v$. Since we know $D_{tot}$ to first order in $\lambda$, we can thus find $X$, and consequently $\widetilde{\mathcal{L}}$, to second order. Evaluating

$$\mathcal{D}'_\lambda D_0 = \frac{w}{R_v}(\Delta\mathcal{O} - wE_R\frac{t}{R_v}) - \frac{w}{R}\int d\sigma \frac{\mathcal{H}_R}{\sqrt{}}\frac{wt + R\hat{\phi}(\sigma)}{R_v} - \frac{E_R}{R}\int d\sigma \frac{\mathcal{J}_- + \lambda\mathcal{H}_R/2}{\sqrt{}}\frac{wt + R\hat{\phi}(\sigma)}{R_v} \tag{3.3}$$

gives us

$$D(t) = \frac{w\chi_0 - E_R\phi_0}{R} - wE_R\frac{t}{R_v} + \frac{\lambda w}{R_v}\int d\sigma \frac{\mathcal{H}_R}{\sqrt{}}\left(\hat{\phi} - \frac{\phi_0}{R}\right) + \frac{\lambda E_R}{R_v}\int d\sigma \frac{\mathcal{J}_- + \lambda\mathcal{H}_R/2}{\sqrt{}}\left(\hat{\phi} - \frac{\phi_0}{R}\right) + \ldots \tag{3.4}$$

It is extremely useful to note that to this order, $D(t)$ can be written as

$$D(t) \approx \frac{w\widetilde{\chi}_0 - E_R\widetilde{\phi}_0}{R} - wE_R\frac{t}{R_v} - \frac{\lambda\phi_0 E_R Q_K}{R^2} + \mathcal{O}(\lambda^2), \tag{3.5}$$

where the "improved" zero modes $\widetilde{\phi}_0$ and $\widetilde{\chi}_0$ are defined as

$$\widetilde{\phi}_0 \equiv \phi_0 - \frac{\lambda R}{R_v}\int d\sigma(\mathcal{J}_- + \lambda\mathcal{H}_R/2)\hat{\phi}, \qquad \widetilde{\chi}_0 \equiv \chi_0 + \frac{\lambda R}{R_v}\int d\sigma \mathcal{H}_R\hat{\phi}. \tag{3.6}$$

The usefulness of introducing these quantities stems from the extremely simple Poisson brackets they satisfy, to *all orders* in $\lambda$. The Poisson brackets of $\widetilde{\phi}_0$ with the left-movers are

$$\{\widetilde{\phi}_0, K_U\} = \frac{1}{2}, \qquad \{\widetilde{\phi}_0, \mathcal{H}_L\} = K_U, \tag{3.7}$$

and with the right-movers

$$\{\widetilde{\phi}_0, \bar{Q}_{\bar{f}}\} = \frac{R}{R_v}\bar{P}^{KM}_{\bar{f}}, \qquad \{\widetilde{\phi}_0, \bar{P}^{KM}_{\bar{\eta}}\} = \frac{R}{2R_v}\int d\sigma\bar{\eta}(1 - \lambda\phi'), \tag{3.8}$$

which implies that $\widetilde{\phi}_0$ is the corrected operator implementing spectral flow in the $J\bar{T}$ - deformed CFT. The field $\widetilde{\chi}_0$ commutes with $K_U, \mathcal{H}_L, E_R, \bar{J}_0$ and, for our particular choice of the constant $a$ in the Poisson brackets, with everything else[6].

It is also interesting to check the flow equations that $\widetilde{\phi}_0$ and $\widetilde{\chi}_0$ satisfy[7]

$$\mathcal{D}'_\lambda \widetilde{\chi}_0 = \frac{w}{R_v} \widetilde{\chi}_0 , \qquad \mathcal{D}'_\lambda \widetilde{\phi}_0 = \frac{wtR}{R_v^2} \bar{Q}_{\bar{K}} . \tag{3.12}$$

Given the rewriting (3.5) of $D(t)$, it is best to consider a different identification of what is considered 'zeroth' versus 'first' order in its expansion with respect to $\lambda$, namely

$$\widetilde{D}_0 = \frac{w\widetilde{\chi}_0 - E_R \widetilde{\phi}_0}{R} - wE_R \frac{t}{R_v} + \mathcal{O}(\lambda^2) , \qquad \widetilde{D}_1 = -\frac{\phi_0 E_R Q_K}{R^2} + \mathcal{O}(\lambda). \tag{3.13}$$

Then, the coefficients of the perturbative expansion (3.1) of $X$, obtained using (2.45), are given by

$$\mathcal{O}_1 = \frac{w\widetilde{\chi}_0 - E_R \widetilde{\phi}_0}{R} - wE_R \frac{t}{R_v} + \frac{2Q_K E_R t}{R} , \qquad \mathcal{D}'_\lambda \mathcal{O}_1 = \frac{w}{R_v} \frac{w\widetilde{\chi}_0 - E_R \widetilde{\phi}_0}{R} + \frac{2Q_K wE_R t}{R^2} - \frac{wtE_R(Q_K + \omega)}{R^2} , \tag{3.14}$$

which implies that, to zeroth order in $\lambda$

$$
\begin{aligned}
R^2 \mathcal{O}_2 &= -\frac{1}{2}\phi_0 E_R Q_K - \frac{w(w\widetilde{\chi}_0 - E_R \widetilde{\phi}_0)}{2} - Q_K wE_R t + Q_K^2 E_R t + \frac{1}{2} wtE_R(Q_K + \omega) \\
&\approx -\frac{1}{2}\phi_0 E_R \bar{Q}_{\bar{K}} - \frac{\omega^2}{2}\chi_0 + E_R Q_K \bar{Q}_{\bar{K}} t + \frac{1}{2} wtE_R(Q_K + \omega).
\end{aligned}
\tag{3.15}
$$

The above expressions for $\mathcal{O}_{1,2}$ give us the classical limit of the operator $X$ entering the similarity transformation, up to $\mathcal{O}(\lambda^3)$.

We would now like to check the effect of the similarity transformation on the various conserved charges. To pass from the quantum commutators to Poisson brackets, we note that the operator $X$ should have a factor of $\hbar^{-1}$ in front, which cancels against the $\hbar$ factors in the commutators to yield the classical result

$$\widetilde{\mathcal{L}} = e^X \mathcal{L} e^{-X} \;\longleftrightarrow\; \widetilde{\mathcal{L}}_{cls} = \mathcal{L} + \lambda\{\mathcal{O}_1, \mathcal{L}\} + \lambda^2\{\mathcal{O}_1, \mathcal{L}\} + \frac{\lambda^2}{2}\{\mathcal{O}_1, \{\mathcal{O}_1, \mathcal{L}\}\} + \mathcal{O}(\lambda^3). \tag{3.16}$$

Let us first work out the effect of this transformation on $K_U$. Using

$$\{\mathcal{O}_1, K_U\} = -\frac{E_R}{2R} , \qquad \{\mathcal{O}_2, K_U\} = -\frac{E_R \bar{Q}_{\bar{K}}}{2R^2} \{\phi_0, K_U\} = -\frac{E_R \bar{Q}_{\bar{K}}}{4R^2} , \qquad \{\mathcal{O}_1, E_R\} = -\frac{E_R \bar{Q}_{\bar{K}}}{R}, \tag{3.17}$$

---

[6]For $a \neq -1$, its non-zero commutators are

$$\{\widetilde{\chi}_0, \mathcal{H}_R\} = (1+a)R\partial_\sigma \frac{\mathcal{H}_R}{\sqrt{}} , \qquad \{\widetilde{\chi}_0, \mathcal{J}_-\} = (1+a)R\partial_\sigma \frac{\mathcal{J}_-}{\sqrt{}} , \qquad \{\widetilde{\chi}_0, \phi\} = -(1+a)R\frac{\mathcal{J}_- + \lambda/2\mathcal{H}_R}{\sqrt{}} , \tag{3.9}$$

which implies

$$\{\widetilde{\chi}_0, \bar{Q}_{\bar{f}}\} = -\frac{R}{R_v}\bar{Q}_{\bar{f}'}(1+a) , \qquad \{\widetilde{\chi}_0, \bar{P}_{\bar{\eta}}^{KM}\} = -\frac{\bar{P}_{\bar{\eta}'}^{KM}R}{R_v}(1+a) , \qquad \{\widetilde{\phi}_0, \widetilde{\chi}_0\} = \frac{R^2 \bar{Q}_{\bar{K}}}{R_v}(1+a). \tag{3.10}$$

[7]More generally,

$$\mathcal{D}_\lambda \widetilde{\chi}_0 = \mathcal{D}'_\lambda \widetilde{\chi}_0 = \frac{w}{R_v}\widetilde{\chi}_0 + \frac{wR}{R_v}E_R(1+a) , \qquad \mathcal{D}_\lambda \widetilde{\phi}_0 = -\frac{R\bar{Q}_{\bar{K}}}{R_v}w(1+a) , \qquad \mathcal{D}'_\lambda \widetilde{\phi}_0 = \frac{wtR}{R_v^2}\bar{Q}_{\bar{K}}. \tag{3.11}$$

we can readily show that

$$\widetilde{K}_U = K_U - \frac{\lambda E_R}{2R} + \mathcal{O}(\lambda^3). \tag{3.18}$$

Note in particular that the zero mode of $\widetilde{K}_U$ is just $J_0$. This implies that the spectrally flowed generators will commute with $J_0$, since the commutator $[J_0, e^X \bar{Q}_{\bar{f}} e^{-X}] = e^X [e^{-X} J_0 e^X, \bar{Q}_{\bar{f}}] e^{-X} = e^X [Q_K, \bar{Q}_{\bar{f}}] e^{-X} = 0$.

Next, we would like to check what happens to the left-moving energy current $\mathcal{H}_L$. We evaluate

$$\{\mathcal{O}_1, \mathcal{H}_L\} = -\frac{E_R K_U}{R}, \qquad \{\mathcal{O}_2, \mathcal{H}_L\} = -\frac{E_R \bar{Q}_{\bar{K}}}{2R^2} \{\phi_0, \mathcal{H}_L\} = -\frac{E_R \bar{Q}_{\bar{K}}}{2R^2} K_U, \tag{3.19}$$

$$\{\mathcal{O}_1, -E_R K_U\} = \frac{E_R^2}{2R} + \frac{E_R K_U \bar{Q}_{\bar{K}}}{R}, \tag{3.20}$$

which in turn implies that

$$\widetilde{\mathcal{H}}_L = \mathcal{H}_L - \frac{\lambda E_R K_U}{R} + \frac{\lambda^2 E_R^2}{4R^2} + \mathcal{O}(\lambda^3). \tag{3.21}$$

Note that the transformation of $K_U$ and that of $\mathcal{H}_L$ correspond precisely to a spectral flow transformation with parameter $\lambda E_R$. The zero mode of $\widetilde{\mathcal{H}}_L$ equals the left-moving energy $E_L^{(0)}$ in the undeformed CFT.

Let us now turn to the right-movers, starting with the Kac-Moody generators $\bar{P}_{\bar{\eta}}^{KM}$ in (1.6). Using (3.8), we compute

$$\{\mathcal{O}_1, \bar{P}_{\bar{\eta}}^{KM}\} = -\frac{E_R}{2R_\nu} \int d\sigma \bar{\eta} (1 - \lambda \phi') + \frac{1}{R_\nu} \left( \frac{\widetilde{\phi}_0 - 2Q_K t}{R} + \frac{wt}{R_\nu} \right) \bar{P}_{\bar{\eta}'}^{KM}, \tag{3.22}$$

$$\{\mathcal{O}_2, \bar{P}_{\bar{\eta}}^{KM}\} \approx -\frac{E_R \bar{Q}_{\bar{K}}}{4R_\nu R} \int d\sigma \bar{\eta} (1 - \lambda \phi') + \frac{\bar{Q}_{\bar{K}}}{2R^2} (\widetilde{\phi}_0 - 2Q_K t) \bar{P}_{\bar{\eta}'}^{KM}, -\frac{wt}{2R^3} (Q_K + \omega) \bar{P}_{\bar{\eta}'}^{KM}. \tag{3.23}$$

It is useful to treat separately the case in which $\eta = I$ (the identity), for which

$$\widetilde{\bar{P}}_I^{KM} = \left( \bar{J}_0 + \frac{\lambda E_R}{2} \right) - \frac{\lambda E_R}{2} - \frac{\lambda^2}{4R} E_R \bar{Q}_{\bar{K}} + \frac{\lambda^2}{2} \{\mathcal{O}_1, -E_R/2\} = \bar{J}_0 + \mathcal{O}(\lambda^3), \tag{3.24}$$

as expected from spectral flow. For $\bar{\eta} \neq I$, we compute

$$R^2 \{\mathcal{O}_1, \{\mathcal{O}_1, \bar{P}_{\bar{\eta}}^{KM}\}\} = \left( \frac{\phi_0 - 2Q_K t}{R} + \frac{wt}{R_\nu} \right) \bar{Q}_{\bar{K}} \bar{P}_{\bar{\eta}'}^{KM} + \left( \frac{\phi_0 - 2Q_K t}{R} + \frac{wt}{R_\nu} \right)^2 \bar{P}_{\bar{\eta}''}^{KM}. \tag{3.25}$$

The transformed $\bar{P}_{\bar{\eta}}^{KM}$ is then, for $\bar{\eta} \neq I$

$$
\begin{aligned}
\widetilde{\bar{P}}_{\bar{\eta}}^{KM} &= \bar{P}_{\bar{\eta}}^{KM} + \frac{\lambda}{R_\nu} \left( \frac{\widetilde{\phi}_0 - 2Q_K t}{R} + \frac{wt}{R_\nu} \right) \bar{P}_{\bar{\eta}'}^{KM} + \frac{\lambda^2}{2R^2} \left( \frac{\widetilde{\phi}_0 - 2Q_K t}{R} + \frac{wt}{R_\nu} \right)^2 \bar{P}_{\bar{\eta}''}^{KM} \\
&\quad + \frac{\lambda^2}{R^3} \left[ \bar{Q}_{\bar{K}} (\phi_0 - 2Q_K t) - \omega^2 t \right] \bar{P}_{\bar{\eta}'}^{KM} + \mathcal{O}(\lambda^3) \\
&\approx \bar{P}_{\bar{\eta}}^{KM} + \hat{\lambda} (1 + \hat{\lambda} Q_K) \left( \frac{\widetilde{\phi}_0 - 2Q_K t + \omega t - \hat{\lambda} \omega \bar{Q}_{\bar{K}} t}{R} \right) \bar{P}_{\bar{\eta}'}^{KM}
\end{aligned}
\tag{3.26, 3.27}
$$

$$+ \frac{\hat{\lambda}^2}{2} \left( \frac{\widetilde{\phi}_0 - 2Q_K t + \omega t}{R} \right)^2 \bar{P}_{\bar{\eta}''}^{KM} + \mathcal{O}(\lambda^3), \tag{3.28}$$

where we have introduced the dimensionless $J\bar{T}$ coupling $\hat{\lambda} = \lambda/R$. It is not hard to check that this expression is conserved to the given order in $\lambda$, using

$$\{H, \widetilde{\phi}_0\} = -Q_K - \frac{\bar{Q}_{\bar{K}} R}{R_\nu}. \tag{3.29}$$

Using $\{P, \widetilde{\phi}_0\} = -Q_K + \frac{\bar{Q}_{\bar{K}} R}{R_\nu} \approx -w(1 - \hat{\lambda}\bar{Q}_{\bar{K}})$ and the approximate expansion above, we also find

$$\{P, \widetilde{\bar{P}}^{KM}_{\tilde{\eta}}\} = \frac{1}{R} \widetilde{\bar{P}}^{KM}_{\tilde{\eta}'} + \mathcal{O}(\lambda^3), \tag{3.30}$$

i.e. acting with the spectrally flowed right-moving generator preserves the quantization of the momentum.

Finally, moving on to the right-moving pseudoconformal generators, we find

$$\{\mathcal{O}_1, \bar{Q}_{\tilde{f}}\} = -\frac{E_R \bar{P}^{KM}_{\tilde{f}}}{R_\nu} + \frac{1}{R_\nu}\left(\frac{\widetilde{\phi}_0 - 2Q_K t}{R} + \frac{wt}{R_\nu}\right)\bar{Q}_{\tilde{f}'} \tag{3.31}$$

$$\{\mathcal{O}_2, \bar{Q}_{\tilde{f}}\} = -\frac{E_R \bar{Q}_{\bar{K}}}{2R_\nu R}\bar{P}^{KM}_{\tilde{f}} + \frac{\bar{Q}_{\bar{K}}}{2R_\nu R^2}(\widetilde{\phi}_0 - 2Q_K t)\bar{Q}_{\tilde{f}'} - \frac{wt(Q_K + \omega)}{2R^3}\bar{Q}_{\tilde{f}'} \tag{3.32}$$

$$\{\mathcal{O}_1, \{\mathcal{O}_1, \bar{Q}_{\tilde{f}}\}\} = -\frac{E_R}{R_\nu}\{\mathcal{O}_1, \bar{P}^{KM}_{\tilde{f}}\} + \frac{E_R \bar{Q}_{\bar{K}}}{R_\nu R}\bar{P}^{KM}_{\tilde{f}} + \frac{\frac{\phi_0 - 2Q_K t}{R} + \frac{wt}{R_\nu}}{R_\nu}(\bar{Q}_{\bar{K}}\bar{Q}_{\tilde{f}'} - E_R \bar{P}^{KM}_{\tilde{f}'}) + \frac{(\frac{\phi_0 - 2Q_K t}{R} + \frac{wt}{R_\nu})^2}{R_\nu^2}\bar{Q}_{\tilde{f}''}. \tag{3.33}$$

Let us first check the case $f = const$. Remembering that the orginal right-moving pseudoconformal generator that satisfies $\mathcal{D}'_\lambda \mathcal{L} = 0$ is $\bar{Q}_{\tilde{f}} R_\nu$, applying the similarity transformation to it yields

$$\widetilde{E}_R R = E_R R_\nu - \lambda E_R \bar{J}_0 - \frac{\lambda^2}{4}E_R^2 + \mathcal{O}(\lambda^3), \tag{3.34}$$

in perfect agreement with our expectation, $RE_R^{(0)}$. The factor of $R$ on the left-hand side has been included for dimensional reasons. For general $f$, we find

$$\begin{aligned}\widetilde{\bar{Q}}_{\tilde{f}} R &= \bar{Q}_{\tilde{f}} R_\nu - \lambda E_R \bar{P}^{KM}_{\tilde{f}} + \frac{\lambda^2 E_R^2}{4}\delta_{f=I} + \lambda\left(\frac{\widetilde{\phi}_0 - 2Q_K t}{R} + \frac{wt}{R_\nu}\right)\bar{Q}_{\tilde{f}'} \\ &+ \frac{\lambda^2}{2R^2}(\phi_0 - 2Q_K t)(2\bar{Q}_{\bar{K}}\bar{Q}_{\tilde{f}'} - 2E_R \bar{P}^{KM}_{\tilde{f}'}) + \frac{\lambda^2}{2R}\left(\frac{\phi_0 - 2Q_K t}{R} + \frac{wt}{R_\nu}\right)^2 \bar{Q}_{\tilde{f}''} \\ &- \frac{\lambda^2 \omega t}{2R^2}(Q_K + \omega)\bar{Q}_{\tilde{f}'} + \frac{\lambda^2 \omega t}{2R^2}(\bar{Q}_{\bar{K}}\bar{Q}_{\tilde{f}'} - 2E_R \bar{P}^{KM}_{\tilde{f}'}). \end{aligned} \tag{3.35}$$

As before, this can be organised as the following perturbative expansion

$$\begin{aligned}\widetilde{\bar{Q}}_{\tilde{f}} R &= R_\nu\left(\bar{Q}_{\tilde{f}} + \hat{\lambda}(1 + \hat{\lambda}Q_K)\frac{\widetilde{\phi}_0 - 2Q_K t + \omega t(1 - \hat{\lambda}\bar{Q}_{\bar{K}})}{R}\bar{Q}_{\tilde{f}'} + \frac{\hat{\lambda}^2}{2R^2}(\phi_0 - 2Q_K t + \omega t)^2 \bar{Q}_{\tilde{f}''}\right) \\ &- \lambda E_R\left(\bar{P}^{KM}_{\tilde{f}} + \frac{\lambda(\phi_0 - 2Q_K t + \omega t)}{R^2}\bar{P}^{KM}_{\tilde{f}'}\right) + \frac{\lambda^2 E_R^2}{4}\delta_{f=I} + \mathcal{O}(\lambda^3). \end{aligned} \tag{3.36}$$

This has precisely the correct form to yield a conserved charge and an integer-quantized momentum, as one can check by computing its Poisson brackets with $H$ and $P$.

## 3.2. An all-orders proposal

The result of the perturbative analysis we have just performed is that the symmetry generators that act properly on the eigenstates of the system are given by a kind of energy-dependent spectral flow. While the form of the resulting left-moving generators, $\widetilde{K}_U$ and $\widetilde{\mathcal{H}}_L$ in (3.18) and (3.21), matches precisely to what we expect from spectral flow with parameter $\lambda E_R$, the form of the right-moving generators (3.28) and (3.36) is significantly more involved. In particular, while it is nothing but natural that (3.18) and (3.21) should represent the full expressions for the left-moving flowed generators to all orders in $\lambda$, it is also clear that the $\lambda$ expansion of the right-moving generators will contain an infinite number of terms.

In this section, we will make a proposal for an all-orders (formal) expression for the right-moving generators, starting from the assumption that (3.18) and (3.21) are the correct expression for the flowed left-moving currents to all orders in $\lambda$. Our main tool will be the fact that the charge algebra is preserved by the flow (2.40), and therefore the spectrally flowed left and right generators should commute to all orders in $\lambda$.

Our analysis will proceed in two steps. First, we will construct combinations of the right-moving conserved charges that commute with the left-moving spectrally flowed currents, and show that these building blocks satisfy the correct Poisson brackets with the energy, momentum and the global $U(1)$ charges to have, upon quantization, a consistent action on the Hilbert space. Then, we find linear combinations of these blocks that satisfy the expected flow equation, with an operator $D$ we will similarly derive.

Let us start by analysing the building block, $\widetilde{\overline{\mathcal{P}}}_{\bar{\eta}}$, for the right-moving $U(1)$ generator. The requirement that it commute with all the left-moving charges (or, alternatively, the currents), reads

$$\left\{K_U - \frac{\lambda}{2R}E_R, \widetilde{\overline{\mathcal{P}}}_{\bar{\eta}}\right\} = 0, \qquad \left\{\mathcal{H}_L - \frac{\lambda E_R K_U}{R} + \frac{\lambda^2}{4R^2}E_R^2, \widetilde{\overline{\mathcal{P}}}_{\bar{\eta}}\right\} = 0, \tag{3.37}$$

where $\bar{\mathcal{P}}_{\bar{\eta}}$ can in fact be *any* right-moving current. Note the second equation follows from the first if

$$\{\mathcal{H}_L, \widetilde{\overline{\mathcal{P}}}_{\bar{\eta}}\} = 2K_U\{K_U, \widetilde{\overline{\mathcal{P}}}_{\bar{\eta}}\}, \tag{3.38}$$

which can *a posteriori* be checked to be the case. Remembering that

$$\{\widetilde{\bar{\phi}}_0, K_U\} = \frac{1}{2}, \qquad \{\widetilde{\bar{\phi}}_0, E_R\} = \frac{\bar{Q}_{\bar{K}}R}{R_v}, \qquad \{E_R, \bar{P}_{\bar{\eta}}\} = -\frac{1}{R_v}\bar{P}_{\bar{\eta}'}, \tag{3.39}$$

a natural Ansatz for $\widetilde{\overline{\mathcal{P}}}_{\bar{\eta}}$ (for $\eta \neq I$) is

$$\widetilde{\overline{\mathcal{P}}}_{\bar{\eta}} = \bar{P}_{\bar{\eta}} + \hat{\lambda}a_1\widetilde{\bar{\phi}}_0\bar{P}_{\bar{\eta}'} + \frac{\hat{\lambda}^2 a_2}{2}\widetilde{\bar{\phi}}_0^2\bar{P}_{\bar{\eta}''} + \dots \tag{3.40}$$

Plugging into (3.37), we find the recursion relation

$$a_{n+1} = \frac{a_n}{R - \lambda Q_K}, \qquad a_0 = 1, \tag{3.41}$$

which implies that the solution is simply

$$\widetilde{\overline{\mathcal{P}}}_{\bar{\eta}} = \bar{P}_{\bar{\eta}} + \frac{\hat{\lambda}\widetilde{\bar{\phi}}_0}{R - \lambda Q_K}\bar{P}_{\bar{\eta}'} + \frac{\hat{\lambda}^2\widetilde{\bar{\phi}}_0^2}{2(R - \lambda Q_K)^2}\bar{P}_{\bar{\eta}'} + \dots \tag{3.42}$$

Note that in the case of the Kac-Moody current $P_{\bar{\eta}}^{KM}$, the first three terms agree precisely with the result (3.28) of the perturbative analysis of the previous section, at $t = 0$. One can also check that (3.38) holds, using (3.7), (A.13) and the commutators of $\phi$ with $\mathcal{H}_L$ and $K_U$.

Note the above is a formal expression in that $\phi_0$, and thus $\widetilde{\phi}_0$, is not a well-defined operator. However, its exponential is expected to be. Using the $\widetilde{\phi}_0$ Poisson brackets

$$\{\widetilde{\phi}_0, J_0\} = \{\widetilde{\phi}_0, \bar{J}_0\} = R\,\frac{R - \lambda Q_K}{2R_\nu}, \tag{3.43}$$

with $Q_K \equiv J_0 + \lambda E_R/2$ and $\bar{Q}_{\bar{K}} \equiv \bar{J}_0 + \lambda E_R/2$ that we presented earlier, we can easily check that the charges of this combination are as expected, namely

$$\{J_0, \widetilde{\mathcal{P}}_{\bar{\eta}}\} = \{\bar{J}_0, \widetilde{\mathcal{P}}_{\bar{\eta}}\} = \frac{\lambda}{2R_\nu}\widetilde{\mathcal{P}}_{\bar{\eta}'} - \frac{(R - \lambda Q_K)}{2R_\nu}\frac{\lambda}{R - \lambda Q_K}\widetilde{\mathcal{P}}_{\bar{\eta}'} = 0. \tag{3.44}$$

The Poisson bracket with the right-moving energy is

$$\{E_R, \widetilde{\mathcal{P}}_{\bar{\eta}}\} = -\frac{\lambda}{R - \lambda Q_K}\frac{\bar{Q}_{\bar{K}}}{R_\nu}\widetilde{\mathcal{P}}_{\bar{\eta}'} - \frac{1}{R_\nu}\widetilde{\mathcal{P}}_{\bar{\eta}'} = -\frac{1}{R - \lambda Q_K}\widetilde{\mathcal{P}}_{\bar{\eta}'}, \tag{3.45}$$

while the commutator with $\mathcal{H}_L$ is given by

$$\{\mathcal{H}_L, \widetilde{\mathcal{P}}_{\bar{\eta}}\} = \frac{\lambda K_U}{R}\{E_R, \widetilde{\mathcal{P}}_{\bar{\eta}}\} = -\frac{\lambda K_U}{R(R - \lambda Q_K)}\widetilde{\mathcal{P}}_{\bar{\eta}'}. \tag{3.46}$$

These together imply that the Poisson bracket with the total momentum is

$$\{P, \widetilde{\mathcal{P}}_{\bar{\eta}}\} = \{E_L - E_R, \widetilde{\mathcal{P}}_{\bar{\eta}}\} = -\frac{1}{R}\widetilde{\mathcal{P}}_{\bar{\eta}'}, \tag{3.47}$$

and thus the action of $\widetilde{\mathcal{P}}_{\bar{\eta}}$ in the Fourier basis increases the momentum by an integer amount in units of the radius, which is now consistent with semiclassical quantization. The total energy is given by

$$\{E_L + E_R, \widetilde{\mathcal{P}}_{\bar{\eta}}\} = -\frac{R + \lambda Q_K}{R - \lambda Q_K}\frac{\widetilde{\mathcal{P}}_{\bar{\eta}'}}{R}. \tag{3.48}$$

To ensure conservation of the charges, one should, for $t \neq 0$, replace $\widetilde{\phi}_0$ by the block

$$\widetilde{\phi}_0 - \left(Q_K + \frac{\bar{Q}_{\bar{K}}R}{R - \lambda w}\right)t, \tag{3.49}$$

which is conserved by itself. This agrees precisely with what happened in our previous perturbative analysis, and makes it manifest that each term in the sum (3.42) is separately conserved.

It is also interesting to note that (3.45) implies that the spectrally flowed right-moving energy $E_R^{(0)}$ is also changed by an integer amount

$$\left\{E_R^{(0)}, \widetilde{\mathcal{P}}_{\bar{\eta}}\right\} = \left\{E_R - \frac{\lambda J_0 E_R}{R} - \frac{\lambda^2 E_R^2}{4R^2}, \widetilde{\mathcal{P}}_{\bar{\eta}}\right\} = \{E_R, \widetilde{\mathcal{P}}_{\bar{\eta}}\}\left(1 - \frac{\lambda Q_K}{R}\right) = -\frac{1}{R}\widetilde{\mathcal{P}}_{\bar{\eta}'}, \tag{3.50}$$

as expected from the fact that it is the global mode $(\widetilde{\bar{L}}_0)$ of the spectrally flowed algebra.

An identical analysis for the case of the pseudoconformal generators shows that they must appear in the combination

$$\widetilde{\mathcal{Q}}_{\bar{f}} = \bar{Q}_{\bar{f}} + \frac{\hat{\lambda}\widetilde{\phi}_0}{R - \lambda Q_K}\bar{Q}_{\bar{f}'} + \frac{\hat{\lambda}^2\widetilde{\phi}_0^2}{2(R - \lambda Q_K)^2}\bar{Q}_{\bar{f}''} + \dots, \tag{3.51}$$

and their commutation relations are exactly analogous with those of $\widetilde{\mathcal{P}}_{\bar{\eta}}$.

As already explained, the formal expressions $\widetilde{\mathcal{P}}_{\bar\eta}$ and $\widetilde{\mathcal{Q}}_{\bar f}$ are not exactly the spectrally flowed $\widetilde{\mathcal{L}}$ generators, as the commutation requirement (3.37) is a weaker condition than the flow equation (2.40). Instead, they represent the building blocks of the spectrally flowed generators. To find which linear combination of them represents the $\widetilde{\mathcal{L}}$, we now turn to the flow equation they satisfy.

It turns out that the flow operator $D$ can also be fixed by its commutation relations with $K_U$ and $\mathcal{H}_L$, upon choosing a judicious Ansatz. Assuming that the spectrally flowed left-moving generators are given precisely by (3.18) and (3.21), the flow equation (2.40) they are expected to satisfy and the fact (C.2) that they are annihilated by $\mathcal{D}_\lambda$ fix the commutation relations of $D$ to to all orders to

$$\{D, K_U - \frac{\lambda}{2R}E_R\} = \mathcal{D}_\lambda(K_U - \frac{\lambda}{2R}E_R) = -\frac{1}{2R_\nu}E_R\,, \quad \{D, \mathcal{H}_L - \frac{\lambda E_R K_U}{R} + \frac{\lambda^2 E_R^2}{4R^2}\} = -\frac{E_R}{R_\nu}(K_U - \frac{\lambda E_R}{2R})\,. \tag{3.52}$$

We make the following Ansatz for $D$

$$D = a(\lambda)\widetilde{\phi}_0 + b(\lambda)\widetilde{\chi}_0 + c(\lambda)\,, \tag{3.53}$$

where the operators $a, b, c$ commute with both $K_U$ and $E_R$. The first equation implies that

$$a - \frac{\lambda}{R}\{D, E_R\} = -\frac{E_R}{R_\nu}\,, \quad \text{where } \{D, E_R\} = \frac{a\bar{Q}_{\bar K}R}{R - \lambda w}\,, \tag{3.54}$$

thus yielding

$$a = -\frac{E_R}{R - \lambda Q_K}\,, \tag{3.55}$$

in perfect agreement with our perturbative solution (3.5). While the coefficient $b$ is not fixed, this is not very important, since with our choice of $\chi_0$ Poisson brackets, $\widetilde{\chi}_0$ commutes with all operators. We will set $b = w/R$ to match with the perturbative answer, and $c = 0$, at least on the $t = 0$ slice.

We would now like to show that

$$\boxed{\widetilde{P}_{\bar\eta}^{KM} \equiv \widetilde{\mathcal{P}}_{\bar\eta} + \frac{\lambda}{2}\widetilde{\mathcal{Q}}_{\bar\eta} - \frac{\lambda E_R}{2}\,\delta_{\bar\eta=I}} \tag{3.56}$$

and

$$\boxed{R\widetilde{Q}_{\bar f} \equiv R_\nu \widetilde{\mathcal{Q}}_{\bar f} - \lambda E_R \widetilde{\mathcal{P}}_{\bar f}^{KM} - \frac{\lambda^2 E_R^2}{4}\,\delta_{\bar f=I}} \tag{3.57}$$

precisely satisfy the flow equation (2.40) for the above choice of $D$. We compute, for $\eta \neq I$ and at $t = 0$

$$\mathcal{D}_\lambda \widetilde{P}_{\bar\eta}^{KM} = \mathcal{D}_\lambda\left(\frac{\hat\lambda\widetilde{\phi}_0}{R - \lambda Q_K}\right)\widetilde{P}_{\bar\eta'}^{KM} = \frac{\widetilde{\phi}_0}{(R - \lambda Q_K)^2}\widetilde{P}_{\bar\eta'}^{KM} = \left\{\frac{w\widetilde{\chi}_0}{R} - \frac{E_R\widetilde{\phi}_0}{R - \lambda Q_K}, \widetilde{P}_{\bar\eta}^{KM}\right\}\,. \tag{3.58}$$

The case $\eta = I$, for which $\widetilde{P}_{\bar I}^{KM} = \bar{J}_0$ needs to be worked out separately, and it is easy to check that

$$\mathcal{D}_\lambda \bar{J}_0 = \{D, \bar{J}_0\}\,, \tag{3.59}$$

which justifies the constant shift by $-\lambda E_R/2$. As far as the pseudoconformal charges are concerned, we find that, for $\bar\eta \neq I$

$$\mathcal{D}'_\lambda \widetilde{\mathcal{Q}}_{\bar\eta} - \left\{\frac{w\widetilde{\chi}_0}{R} - \frac{E_R\widetilde{\phi}_0}{R - \lambda Q_K}, \widetilde{\mathcal{Q}}_{\bar\eta}\right\} = \frac{\omega}{R_\nu}\widetilde{\mathcal{Q}}_{\bar\eta} + \frac{R E_R}{R_\nu(R - \lambda Q_K)}\widetilde{\mathcal{P}}_{\bar\eta}^{KM}\,. \tag{3.60}$$

We can now easily show that the linear combination (3.57) is exactly annihilated by $\mathcal{D}_\lambda - D$. The constant shift follows from the fact that $\{\mathcal{D}_\lambda - D, E_R\} = E_R Q_K / (R - \lambda Q_K)$, which implies in particular that

$$\left\{ \mathcal{D}_\lambda - D, E_R R_\nu - \lambda E_R \bar{J}_0 - \frac{\lambda^2}{4} E_R^2 \right\} = 0. \tag{3.61}$$

The equations (3.56) and (3.57) encode the effect of the spectral flow by an amount proportional to the right-moving energy on the right-moving charges. Due to the flow equation they obey, the spectrally flowed generators satisfy the same algebra as at $\lambda = 0$, i.e. two commuting copies of the Witt-Kac-Moody algebra. The charges, defined at $t \neq 0$ via the replacement (3.49), are conserved. The generators of the global part of the algebra are given by

$$\widetilde{Q}_{f=I} = E_L^{(0)}, \qquad \widetilde{\bar{Q}}_{\bar{f}=I} = E_R^{(0)}, \tag{3.62}$$

whose sum *does not* correspond to the energy operator; rather, it is non-linearly related to it via (1.2). This implies that while the Fourier modes of $\widetilde{Q}$ and $\widetilde{P}$ are integer-spaced with respect to $E_L^{(0)}$ and $E_R^{(0)}$ (as follows from the algebra), they have nontrivial commutation relations with $E_L$ and $E_R$, of the form (3.45), (3.46). This resolves the tension mentioned at the beginning of this article, between the integer spacing of the Virasoro descendants and the non-linear dependence (1.1) of the $J\bar{T}$ - deformed energy spectrum on the initial energy.

# 4. Discussion

In this article, we have constructed a new, infinite set of symmetry generators in $J\bar{T}$ - deformed CFTs. Compared to the conserved charges found in [19], these generators still form two commuting copies of the Witt-Kac-Moody algebra, but now their Poisson brackets with the global $U(1)$ charge and the momentum are consistent with semiclassical quantization. It is therefore expected that in the quantum $J\bar{T}$ - deformed CFT on a cylinder, these generators will be acting properly on the Hilbert space of the theory, and thus the spectrum will organise into Virasoro - Kac-Moody representations.

The new symmetry generators are related to those of [19] by a type of energy-dependent spectral flow transformation, whose action on the various generators is given in (3.18), (3.21), (3.56) and (3.57). The expressions (3.42) and (3.51) for the building blocks are somewhat formal and should be resummed in order to make sense. While the descendants obtained via the action of these symmetry generators will have integer-spaced spectrally flowed energies $E_{L,R}^{(0)}$, there is no tension with the known formula (1.1) for the deformed energies, as the latter are measured with respect to the non-flowed generators.

As is clear from (1.1), at large initial right-moving energy, the finite-size spectrum of $J\bar{T}$ - deformed CFTs will become complex. While this is only a problem for the theory on the cylinder, and not on the plane [23], it is interesting to remark that even on the cylinder, this problem is invisible at the level of the algebra and the spectrum of the flowed generators, which simply coincide with those of the undeformed CFT. Rather, the complex energies arise solely as a result of the expression (A.1) for the Hamiltonian, and signal the breakdown of this square root formula. This suggests that the states of the deformed CFT on a cylinder may be well-behaved, and the only problem is with the operator we use to measure the energy (which, however, is also the one used to evolve the system).

There are several interesting future directions. First, one should be able to pass from the classical construction of the symmetry generators to the quantum one by including the appropriate normal ordering prescriptions and check, at least perturbatively, whether their quantum commutators work out as expected. A particularly interesting issue is that of the

central extension, given that the classical Witt algebra is naturally expected to become Virasoro at the quantum level. The holographic analysis of [21] indicates a field-dependent central extension of the form $c(R/R_v)^2$ for the generators defined in [19], where $c$ is the central charge of the original CFT. Given that the spectrally flowed generators are rescaled by a factor of $R_v$ with respect to the original ones suggests that the central extension in the flowed sector has a chance of being identical to that of the original CFT, in agreement with the fact that the $J\bar{T}$ deformation does not change the number of states. It would be very interesting verify this, at least perturbatively.

Other relevant questions are to understand the physical meaning of the spectrally flowed generators, and whether there is a geometric symmetry that they implement. It would also be interesting to see whether the new symmetry generators may have applications also in the deformed CFT on the plane, e.g. in the construction of more complicated observables such as correlation functions. Finally, it would be interesting to understand the role and physical interpretation of the improved zero mode $\widetilde{\chi}_0$, whose action on the Hilbert space is surprisingly minimal.

### Acknowledgements

MG would like to thank Ruben Monten for collaboration on related subjects. This research was supported in part by the ERC Starting Grant 679278 Emergent-BH and the Swedish Research Council grant number 2015-05333.

# A. Summary of Poisson brackets in $J\bar{T}$ - deformed CFTs

In this appendix, we collect the results of [19] on the Poisson brackets of the various currents in the $J\bar{T}$ - deformed CFTs. These were computed using the expression they derived by solving the flow equation for the deformed right-moving Hamiltonian density $\mathcal{H}_R$

$$\mathcal{H}_R = \frac{2}{\lambda^2}\left(1 - \lambda\mathcal{J}_+ - \sqrt{(1-\lambda\mathcal{J}_+)^2 - \lambda^2\mathcal{H}_R^{(0)}}\right), \tag{A.1}$$

in terms of the undeformed one, $\mathcal{H}_R^{(0)}$, together with the commutation relations of the undeformed currents, $\mathcal{H}_R^{(0)}, \mathcal{J}_\pm$ and $\mathcal{P}$. The currents $\mathcal{J}_\pm$ represent the time components of the linear combinations $(J_\alpha \pm \tilde{J}_\alpha)/2$, where $J$ is represented as a $U(1)$ shift current for a scalar field $\phi$, and $\tilde{J} = \star d\phi$ is the corresponding topologically conserved current. In Hamiltonian language

$$\mathcal{J}_\pm = \frac{\pi \pm \phi'}{2}, \tag{A.2}$$

where $\pi$ is the momentum conjugate to $\phi$. Note the expression for $\mathcal{H}_R$ is symmetric under $\pi \leftrightarrow \phi'$, since it only depends on $\mathcal{J}_+$. From this, we conclude that the $J\bar{T}$ and $\tilde{J}\bar{T}$ deformations, which differ by precisely this exchange, lead to the same deformed theory, at least at the classical level.

The commutation relations derived in [19] are

$$\{\mathcal{H}_R(\sigma), \mathcal{H}_R(\tilde{\sigma})\} = -\left(\frac{\mathcal{H}_R(\sigma)}{\sqrt{(1-\lambda\mathcal{J}_+(\sigma))^2 - \lambda^2\mathcal{H}_R^{(0)}(\sigma)}} + \frac{\mathcal{H}_R(\tilde{\sigma})}{\sqrt{(1-\lambda\mathcal{J}_+(\tilde{\sigma}))^2 - \lambda^2\mathcal{H}_R^{(0)}(\tilde{\sigma})}}\right)\partial_\sigma\delta(\sigma-\tilde{\sigma}), \tag{A.3}$$

$$\{\mathcal{P}(\sigma), \mathcal{H}_R(\tilde{\sigma})\} = \left(\mathcal{H}_R(\sigma) + \frac{\mathcal{H}_R(\tilde{\sigma})}{\sqrt{(1-\lambda\mathcal{J}_+(\tilde{\sigma}))^2 - \lambda^2\mathcal{H}_R^{(0)}(\tilde{\sigma})}}\right)\partial_\sigma\delta(\sigma-\tilde{\sigma}), \tag{A.4}$$

$$\{\mathcal{H}_R(\sigma), \mathcal{J}_+(\tilde{\sigma})\} = \frac{\lambda \mathcal{H}_R(\sigma)}{2\sqrt{(1 - \lambda \mathcal{J}_+(\sigma))^2 - \lambda^2 \mathcal{H}_R^{(0)}}} \partial_\sigma \delta(\sigma - \tilde{\sigma}), \tag{A.5}$$

$$\{\mathcal{H}_R(\sigma), \mathcal{J}_-(\tilde{\sigma})\} = -\frac{\mathcal{J}_-(\sigma)}{\sqrt{(1 - \lambda \mathcal{J}_+(\sigma))^2 - \lambda^2 \mathcal{H}_R^{(0)}(\sigma)}} \partial_\sigma \delta(\sigma - \tilde{\sigma}). \tag{A.6}$$

Using the fact that $\mathcal{J}_+ - \mathcal{J}_- = \phi'$ and that the right-hand side of the last two commutators are total $\tilde{\sigma}$ derivatives, we can deduce the commutator of $\mathcal{H}_R$ with $\phi$

$$\{\mathcal{H}_R(\sigma), \phi(\tilde{\sigma})\} = \frac{-\mathcal{J}_-(\sigma) - \lambda \mathcal{H}_R(\sigma)/2}{\sqrt{(1 - \lambda \mathcal{J}_+(\sigma))^2 - \lambda^2 \mathcal{H}_R^{(0)}(\sigma)}} \delta(\sigma - \tilde{\sigma}). \tag{A.7}$$

Since this commutator is obtained by integration, in principle we could add an integration function of $\sigma$ to the right-hand side; however, such an addition would be quite unnatural, given that the commutator is local (i.e., proportional to a $\delta$ function).

The commutators of the momentum and the currents are the same as in the undeformed CFT

$$\{\mathcal{P}(\sigma), \mathcal{P}(\tilde{\sigma})\} = (\mathcal{P}(\sigma) + \mathcal{P}(\tilde{\sigma})) \partial_\sigma \delta(\sigma - \tilde{\sigma}), \qquad \{\mathcal{J}_\pm(\sigma), \mathcal{J}_\pm(\tilde{\sigma})\} = \pm\frac{1}{2} \partial_\sigma \delta(\sigma - \tilde{\sigma}), \tag{A.8}$$

$$\{\mathcal{P}(\sigma), \mathcal{J}_+(\tilde{\sigma})\} = \mathcal{J}_+(\sigma) \partial_\sigma \delta(\sigma - \tilde{\sigma}). \tag{A.9}$$

From here, one can deduce that

$$\{\mathcal{P}(\sigma), \phi(\tilde{\sigma})\} = -\phi'(\sigma) \delta(\sigma - \tilde{\sigma}), \qquad \{\mathcal{J}_\pm(\sigma), \phi(\tilde{\sigma})\} = -\frac{1}{2} \delta(\sigma - \tilde{\sigma}). \tag{A.10}$$

We note that the zero modes $J_0 = \int d\sigma \mathcal{J}_+$, $\bar{J}_0 = \int d\sigma \mathcal{J}_-$ commute with all the other currents in the theory, and their only non-zero commutator is with $\phi$. The winding charge $w = J_0 - \bar{J}_0$ also commutes with $\phi$. This implies in particular that the field-dependent radius $R_v$ commutes with all the operators we consider.

Finally, one can work out the commutators of the chiral current $K_U = \mathcal{J}_+ + \frac{\lambda}{2} \mathcal{H}_R$ and of the left-moving Hamiltonian $\mathcal{H}_L = \mathcal{H}_R + \mathcal{P}$, which take the very simple form

$$\{K_U(\sigma), K_U(\tilde{\sigma})\} = \frac{1}{2} \partial_\sigma \delta(\sigma - \tilde{\sigma}), \qquad \{\mathcal{H}_L(\sigma), K_U(\tilde{\sigma})\} = K_U(\sigma) \partial_\sigma \delta(\sigma - \tilde{\sigma}), \tag{A.11}$$

$$\{\mathcal{H}_L(\sigma), \mathcal{H}_L(\tilde{\sigma})\} = (\mathcal{H}_L(\sigma) + \mathcal{H}_L(\tilde{\sigma})) \partial_\sigma \delta(\sigma - \tilde{\sigma}), \tag{A.12}$$

and their commutators with $\mathcal{H}_R$ are

$$\{\mathcal{H}_R(\sigma), K_U(\tilde{\sigma})\} = -\frac{\lambda \tilde{\mathcal{H}}_R}{2\sqrt{\cdot}} \delta', \qquad \{\mathcal{H}_R(\sigma), \mathcal{H}_L(\tilde{\sigma})\} = \left( \tilde{\mathcal{H}}_R - \frac{\tilde{\mathcal{H}}_R}{\sqrt{\cdot}} \right) \delta', \tag{A.13}$$

which are total $\sigma$ derivatives. In particular, this implies that $\{E_R, K_U\} = \{E_R, \mathcal{H}_L\} = 0$.

## B. Poisson brackets of the non-local field $\chi$

In this appendix, we derive the commutators of the non-local field $\chi$, defined through $\partial_\sigma \chi = \mathcal{H}_R$, with the various other fields in the theory.

The Poisson brackets of $\chi$ are obtained by integrating the corresponding commutators of $\mathcal{H}_R$. Two of these Poisson brackets, namely

$$\{\chi(\sigma), K_U(\tilde{\sigma})\} = -\frac{\lambda\tilde{\mathcal{H}}_R}{2\sqrt{}}\delta, \quad \{\chi(\sigma), \mathcal{H}_L(\tilde{\sigma})\} = \left(\tilde{\mathcal{H}}_R - \frac{\tilde{\mathcal{H}}_R}{\sqrt{}}\right)\delta \tag{B.1}$$

are local, being proportional to $\delta$ functions, and thus we do not include an integration function. Other commutators, however, are significantly more involved, and require working out the consistency conditions imposed by the various Jacobi identities that they satisfy.

In the following, we will frequently use the notation $\sqrt{} = \sqrt{(1-\lambda\mathcal{J}_+(\sigma))^2 - \lambda^2\mathcal{H}_R^{(0)}(\sigma)}$ and $\sqrt{}' = \sqrt{(1-\lambda\mathcal{J}_+(\tilde{\sigma}))^2 - \lambda^2\mathcal{H}_R^{(0)}(\tilde{\sigma})}$, as well as the abbreviations $\tilde{A} = A(\tilde{\sigma})$ and $\delta' = \partial_\sigma\delta(\sigma-\tilde{\sigma})$.

## B.1. Poisson bracket of $\chi$ with $\mathcal{H}_R$

The $\{\chi, \tilde{\mathcal{H}}_R\}$ Poisson bracket is given by integrating the $\{\mathcal{H}_R, \tilde{\mathcal{H}}_R\}$ commutator

$$\{\chi(\sigma), \mathcal{H}_R(\tilde{\sigma})\} = -\frac{2\mathcal{H}_R}{\sqrt{}}\delta(\sigma-\tilde{\sigma}) + \partial_{\tilde{\sigma}}\frac{\tilde{\mathcal{H}}_R}{\sqrt{}'}\Theta(\sigma-\tilde{\sigma}) + A(\tilde{\sigma}), \tag{B.2}$$

where $A(\tilde{\sigma})$ is an integration function. This function needs to have winding, in order to cancel the dependence on the starting point in

$$\{\chi(\sigma), H_R\} = -\frac{\mathcal{H}_R}{\sqrt{}} - \frac{\mathcal{H}_R(0)}{\sqrt{}(0)} + \int_0^R d\tilde{\sigma} A(\tilde{\sigma}) \equiv -\frac{\mathcal{H}_R}{\sqrt{}} + \int_0^R d\tilde{\sigma} A_p(\tilde{\sigma}), \tag{B.3}$$

where $E_R = \int \mathcal{H}_R d\sigma$ is the total right-moving energy. This equation defines $A_p$, the periodic part of $A$, with the winding subtracted. To fix this function, we need to analyse the constraints coming from the various Jacobi identities that the Poisson bracket (B.2) satisfies.

**Constraints from time evolution**

A first constraint on $A$ comes from analysing the time dependence of the above commutator

$$\frac{d}{dt}\{\chi(\sigma), \mathcal{H}_R(\tilde{\sigma})\} = \partial_t A(\tilde{\sigma}) - \{H, \{\chi(\sigma), \mathcal{H}_R(\tilde{\sigma})\}\} = -\{\{H, \chi\}, \tilde{\mathcal{H}}_R\} - \{\chi, \{H, \tilde{\mathcal{H}}_R\}\}, \tag{B.4}$$

where $H = E_L + E_R$ is the total Hamiltonian. Making use of

$$\{H, \mathcal{H}_R\} = \partial_\sigma(2\mathcal{H}_R/\sqrt{} - \mathcal{H}_R), \quad \{H, \mathcal{J}_+\} = -\partial_\sigma(\mathcal{J}_+ + \lambda\mathcal{H}_R/\sqrt{}), \tag{B.5}$$

$$\{H, \mathcal{H}_R/\sqrt{}\} = \frac{1+\lambda K_U}{1-\lambda K_U}\partial_\sigma(\mathcal{H}_R/\sqrt{}), \tag{B.6}$$

it can be easily shown that the terms proportional to $\Theta, \delta'$ and $\delta$ functions cancel, and the constraint that we obtain on the function $A$ is

$$\partial_t A - \{H, A\} = \int_0^R d\tilde{\sigma}\{A_p(\tilde{\sigma}), \mathcal{H}_R\} - \partial_\sigma\left(\frac{1+\lambda K_U}{1-\lambda K_U}A\right), \tag{B.7}$$

where $A_p$ represents the part of $A$ without the winding contribution. If we choose $A$ such that this term is absent, we find several qualitatively different solutions to the remaining equation. For example,

$$RA \equiv \partial_\sigma\left(\sigma\frac{\mathcal{H}_R}{\sqrt{}}\right) + R\partial_\sigma\hat{A} = \sigma\partial_\sigma\frac{\mathcal{H}_R}{\sqrt{}} + \frac{\mathcal{H}_R}{\sqrt{}} + Ra\,\partial_\sigma\frac{\mathcal{H}_R}{\sqrt{}} \tag{B.8}$$

solves the equations for an arbitrary constant $a$. Another solution can be obtained by noting that the field-dependent coordinate $v = \sigma - t - \lambda\phi$ satisfies

$$\frac{dv}{dt} = -1 - \{H, v\} = -\frac{1 + \lambda K_U}{1 - \lambda K_U} v', \tag{B.9}$$

and thus a general solution to the equation with $A_p = 0$ and the correct winding is

$$A = \partial_\sigma \left( \frac{v}{R_v} \frac{\mathcal{H}_R}{\sqrt{}} \right) + a\, \partial_\sigma \frac{\mathcal{H}_R}{\sqrt{}}, \tag{B.10}$$

where the total derivative is necessary in order to ensure that $A_p = 0$ above.

**Constraints from the Jacobi identity with $K_U$**

To distinguish between the two solutions, we can check the Jacobi identity for $\{K_U, \{\chi_0, \tilde{\mathcal{H}}_R\}\}$. Using the expression for the $\{\chi, \tilde{K}_U\}$ commutator, we obtain

$$\{K_U, (R - \tilde{\sigma})\partial_{\tilde{\sigma}} \frac{\tilde{\mathcal{H}}_R}{\sqrt{}} + R\tilde{A}\} + \frac{\lambda\delta'}{2\sqrt{}} \left[ (R - \sigma)\partial_\sigma \frac{\mathcal{H}_R}{\sqrt{}} + RA \right] = \frac{\lambda}{2\sqrt{}} \partial_\sigma \frac{\mathcal{H}_R}{\sqrt{}} \delta(\sigma - \tilde{\sigma}). \tag{B.11}$$

Taking into account the fact that[8]

$$\{K_U, \frac{\tilde{\mathcal{H}}_R}{\sqrt{}}\} = \frac{\lambda}{2\sqrt{}} \partial_\sigma \frac{\mathcal{H}_R}{\sqrt{}} \delta(\sigma - \tilde{\sigma}), \tag{B.12}$$

we can reduce this to the following simple constraint on $A$

$$\{K_U, \tilde{A}\} + \frac{\lambda}{2\sqrt{}} A\delta' = 0. \tag{B.13}$$

It is easy to check, using (B.12), that the first Ansatz, (B.8), does *not* satisfy the consistency requirement (B.13). On the other hand, using the commutator

$$\{K_U, \tilde{v} \frac{\tilde{\mathcal{H}}_R}{\sqrt{}}\} = -\lambda \frac{\tilde{\mathcal{H}}_R}{\sqrt{}} \{K_U, \tilde{\phi}\} + \tilde{v}\{K_U, \frac{\tilde{\mathcal{H}}_R}{\sqrt{}}\} = \frac{\lambda}{2\sqrt{}} \partial_\sigma(v\mathcal{H}_R/\sqrt{})\delta, \tag{B.14}$$

we can show that the second Ansatz does identically satisfy the consistency condition[9]. One can also check that the term $a\partial_\sigma \mathcal{H}_R/\sqrt{}$ also satisfies it, so this does not fix $a$.

**Constraints from the Jacobi identity with $\mathcal{H}_R$**

We now look at the $\{\chi_0, \mathcal{H}_R\}$ Jacobi identity

$$\{\mathcal{H}_R, \{\chi_0, \tilde{\mathcal{H}}_R\}\} - \{\tilde{\mathcal{H}}_R, \{\chi_0, \mathcal{H}_R\}\} + \{\chi_0, \{\tilde{\mathcal{H}}_R, \mathcal{H}_R\}\} = 0. \tag{B.16}$$

We will also need

$$\{\mathcal{H}_R, \frac{\tilde{\mathcal{H}}_R}{\sqrt{}}\} = -\frac{2\mathcal{H}_R^{(0)}}{\sqrt{}^3} \delta'(\sigma - \tilde{\sigma}) + \left( \frac{1}{\sqrt{}} \partial_\sigma \frac{\mathcal{H}_R}{\sqrt{}} - 2\,\partial_\sigma \frac{\mathcal{H}_R^{(0)}}{\sqrt{}^3} \right) \delta(\sigma - \tilde{\sigma}), \tag{B.17}$$

---

[8]This is most simply computed by using $\sqrt{} = 1 - \lambda K_U$ and the equivalence of distributions spelled out in footnote 10.

[9]Making use (or not) of the relation

$$\{K_U, \tilde{\tilde{\phi}} \frac{\tilde{\mathcal{H}}_R}{\sqrt{}}\} = \frac{\lambda\delta(\sigma - \tilde{\sigma})}{2\sqrt{}} \partial_\sigma \left( \hat{\phi} \frac{\mathcal{H}_R}{\sqrt{}} \right) - \frac{R_v \mathcal{H}_R \delta}{2R(\sqrt{})^2} \tag{B.15}$$

where, from (A.1), $\mathcal{H}_R^{(0)} = \mathcal{H}_R(1 - \lambda \mathcal{J}_+ - \frac{\lambda^2}{4}\mathcal{H}_R)$. Plugging in the expression for

$$\{\chi_0, \mathcal{H}_R\} = -2\mathcal{H}_R/\sqrt{} + (R - \sigma)\partial_\sigma(\mathcal{H}_R/\sqrt{}) + RA, \tag{B.18}$$

we find the intermediate equation

$$R\left(\{\mathcal{H}_R, \tilde{A}\} - \{\tilde{\mathcal{H}}_R, A\} + \frac{A\delta'}{\sqrt{}} + \frac{\tilde{A}\delta'}{\sqrt{\tilde{}}}\right) + (R - \tilde{\sigma})\{\mathcal{H}_R, \partial_{\tilde{\sigma}}\frac{\tilde{\mathcal{H}}_R}{\sqrt{\tilde{}}}\} - (R - \sigma)\{\tilde{\mathcal{H}}_R, \partial_\sigma \frac{\mathcal{H}_R}{\sqrt{}}\}$$
$$+ \left(\frac{R - \sigma}{\sqrt{}}\partial_\sigma \frac{\mathcal{H}_R}{\sqrt{}} + \frac{2\mathcal{H}_R^{(0)}}{\sqrt{}^3} + (\sigma \to \tilde{\sigma})\right)\delta' = 0. \tag{B.19}$$

The difference

$$(R - \tilde{\sigma})\{\mathcal{H}_R, \partial_{\tilde{\sigma}}\frac{\tilde{\mathcal{H}}_R}{\sqrt{\tilde{}}}\} - (R - \sigma)\{\tilde{\mathcal{H}}_R, \partial_\sigma \frac{\mathcal{H}_R}{\sqrt{}}\} \tag{B.20}$$

can be manipulated using the criteria for when two distributions of the form $E(\sigma, \tilde{\sigma})\delta'' + F(\sigma, \tilde{\sigma})\delta' + G(\sigma)\delta$ are equivalent[10]. At the end of the day, we find the very simple constraint

$$\{\mathcal{H}_R, \tilde{A}\} - \{\tilde{\mathcal{H}}_R, A\} + \frac{A\delta'}{\sqrt{}} + \frac{\tilde{A}\delta'}{\sqrt{\tilde{}}} = 0. \tag{B.23}$$

Let us now write $A = A_0 + \hat{A}$, where $A_0 \equiv \partial_\sigma\left(\frac{v}{R_v}\frac{\mathcal{H}_R}{\sqrt{}}\right)$. Evaluating

$$R_v\left(\{\mathcal{H}_R, A_0\} + \frac{A_0\delta'}{\sqrt{}}\right) = \frac{2v\mathcal{H}_R^{(0)}}{\sqrt{}^3}\delta'' + \frac{\mathcal{H}_R}{\sqrt{}}\delta' + 2\partial_\sigma\left(v\frac{\mathcal{H}_R^{(0)}}{\sqrt{}^3}\right)\delta', \tag{B.24}$$

the constraint we obtain on $\hat{A}$ is then simply

$$\{\mathcal{H}_R, \tilde{\hat{A}}\} - \{\tilde{\mathcal{H}}_R, \hat{A}\} + \frac{\hat{A}\delta'}{\sqrt{}} + \frac{\tilde{\hat{A}}\delta'}{\sqrt{\tilde{}}} = -\frac{1}{R_v}\left(\frac{\mathcal{H}_R}{\sqrt{}} + \frac{\tilde{\mathcal{H}}_R}{\sqrt{\tilde{}}}\right)\delta'. \tag{B.25}$$

This is solved by $\hat{A} = \mathcal{H}_R/R_v + \ldots$, where the $\ldots$ are periodic solutions to the homogenous equation, such as the $a\partial_\sigma \mathcal{H}_R/\sqrt{}$ term.

Note however that for this value of $\hat{A}$, we need to revisit the conservation equation (B.7), which receives a new contribution from

$$\int d\tilde{\sigma}\{A_p(\tilde{\sigma}, \mathcal{H}_R(\sigma)\} = \frac{1}{R_v}\{E_R, \mathcal{H}_R\} = \frac{1}{R_v}\partial_\sigma \frac{\mathcal{H}_R}{\sqrt{}}. \tag{B.26}$$

This contribution is very easy to cancel by including an explicit time-dependent term, $t/R_v \partial_\sigma \mathcal{H}_R/\sqrt{}$. This term is also consistent with the $K_U$ and $\mathcal{H}_R$ Jacobi identities. Therefore, the final solution we find for the integration function $A$ is

$$\boxed{A = \partial_\sigma\left[\left(\frac{v}{R_v} + a\right)\frac{\mathcal{H}_R}{\sqrt{}}\right] + \frac{1}{R_v}\left(\mathcal{H}_R + t\partial_\sigma \frac{\mathcal{H}_R}{\sqrt{}}\right) = \partial_\sigma\left[\left(\frac{v + t}{R_v} + a\right)\frac{\mathcal{H}_R}{\sqrt{}}\right] + \frac{\mathcal{H}_R}{R_v},} \tag{B.27}$$

which has the nice feature of not being explicitly time-dependent, since $v + t = \sigma - \lambda\phi$.

---

[10]These crieria are obtained by integrating against a test function $g(\tilde{\sigma})$, with the result

$$g''E + g'(2\partial_{\tilde{\sigma}}E + F) + g(\partial_{\tilde{\sigma}}^2 E + \partial_{\tilde{\sigma}}F + G), \tag{B.21}$$

and respectively $f(\sigma)$, for which

$$f''E + f'(2\partial_\sigma E - F) + f(\partial_\sigma^2 E - \partial_\sigma F + G). \tag{B.22}$$

Two distributions are equivalent if the terms multiplying $f$, $g$ and their various derivatives are the same .

## B.2. Poisson bracket of $\chi$ with $\mathcal{J}_-$

Another commutator that requires special attention is that of $\chi$ with $\mathcal{J}_-$. Integrating the $\{\mathcal{H}_R, \tilde{\mathcal{J}}_-\}$ commutator we obtain

$$\{\chi, \tilde{\mathcal{J}}_-\} = -\frac{\mathcal{J}_-}{\sqrt{}}\delta(\sigma - \tilde{\sigma}) + \partial_{\tilde{\sigma}}\frac{\tilde{\mathcal{J}}_-}{\sqrt{}}\Theta(\sigma - \tilde{\sigma}) + B(\tilde{\sigma}). \tag{B.28}$$

In order for the commutator with $\bar{J}_0$ to be independent of the starting point of the interval, we need the winding of $B$ to equal $\mathcal{J}_-(0)/\sqrt{}(0)$. $B$ should also satisfy all the relevant Jacobi identities.

As before, we first look at the time derivative of this commutator, and try to fix $B$ by requiring that the Jacobi identity hold. We make use of

$$\{H, \mathcal{J}_-/\sqrt{}\} = \frac{1 + \lambda K_U}{1 - \lambda K_U}\partial_\sigma\frac{\mathcal{J}_-}{\sqrt{}} \tag{B.29}$$

and find that $B$ satisfies

$$\partial_t B - \{H, B\} = -\partial_\sigma\left(\frac{1 + \lambda K_U}{1 - \lambda K_U}B\right) + \frac{1}{R_v}\partial_\sigma\frac{\mathcal{J}_-}{\sqrt{}}. \tag{B.30}$$

There are several solutions to this equation that have the correct winding, such as

$$B(\sigma) = \partial_\sigma\left(\frac{\sigma}{R}\frac{\mathcal{J}_-}{\sqrt{}}\right) + b\,\partial_\sigma\frac{\mathcal{J}_-}{\sqrt{}} + \frac{t}{R_v}\partial_\sigma\frac{\mathcal{J}_-}{\sqrt{}} \tag{B.31}$$

or

$$B(\sigma) = \partial_\sigma\left(\frac{v}{R_v}\frac{\mathcal{J}_-}{\sqrt{}}\right) + b\,\partial_\sigma\frac{\mathcal{J}_-}{\sqrt{}} + \frac{t}{R_v}\partial_\sigma\frac{\mathcal{J}_-}{\sqrt{}}, \tag{B.32}$$

for some constant $b$. To find which solution is correct, we need to analyse some further Jacobi identities.

### Constraint from the commutator with $K_U$

We first check the Jacobi identity for $\{K_U, \{\chi_0, \tilde{\mathcal{J}}_-\}\}$. The consistency condition we obtain is

$$\{K_U, (R - \tilde{\sigma})\partial_{\tilde{\sigma}}\frac{\tilde{\mathcal{J}}_-}{\sqrt{}} + R\tilde{B}\} + \frac{\lambda\delta'}{2\sqrt{}}\left[(R - \sigma)\partial_\sigma\frac{\mathcal{J}_-}{\sqrt{}} + RB\right] = \frac{\lambda}{2\sqrt{}}\partial_\sigma\frac{\mathcal{J}_-}{\sqrt{}}\delta(\sigma - \tilde{\sigma}). \tag{B.33}$$

Using the fact that

$$\{K_U, \frac{\tilde{\mathcal{J}}_-}{\sqrt{}}\} = \frac{\lambda}{2\sqrt{}}\partial_\sigma\frac{\mathcal{J}_-}{\sqrt{}}\delta(\sigma - \tilde{\sigma}) \tag{B.34}$$

and the criterion for the equivalence of two distributions, we can reduce the above equation to

$$\{K_U, \tilde{B}\} + \frac{\lambda\delta'}{2\sqrt{}}B = 0. \tag{B.35}$$

It is easy to see that the terms proportional to $\partial_\sigma\mathcal{J}_-/\sqrt{}$ simply drop out of this equation. We then check that the first Ansatz does not solve this equation, whereas the second one does.

**Constraint from the $\mathcal{H}_R$ commutator**

The Jacobi identity reads

$$\{\mathcal{H}_R, \{\chi_0, \tilde{\mathcal{J}}_-\}\} + \{\{\chi_0, \mathcal{H}_R\}, \tilde{\mathcal{J}}_-\} - \{\chi_0, \{\mathcal{H}_R, \tilde{\mathcal{J}}_-\}\} = 0, \tag{B.36}$$

and from its form it is easy to see it will relate the two integration functions $A$ and $B$. To simplify the constraint, we use

$$\{\mathcal{H}_R, \frac{\tilde{\mathcal{J}}_-}{\sqrt{}}\} = -\frac{\mathcal{J}_-(1-\lambda\mathcal{J}_+)}{\sqrt{}^3}\delta' + \left(\frac{1}{\sqrt{}}\partial_\sigma \frac{\mathcal{J}_-}{\sqrt{}} - \partial_\sigma \frac{\mathcal{J}_-(1-\lambda\mathcal{J}_+)}{\sqrt{}^3}\right)\delta(\sigma-\tilde{\sigma}), \tag{B.37}$$

$$\{\frac{\mathcal{H}_R}{\sqrt{}}, \tilde{\mathcal{J}}_-\} = -\frac{\mathcal{J}_-(1-\lambda\mathcal{J}_+)}{\sqrt{}^3}\delta', \tag{B.38}$$

and find that $A$ and $B$ must satisfy the simple relation

$$\{\mathcal{H}_R, \tilde{B}\} + \{A, \tilde{\mathcal{J}}_-\} + \frac{B}{\sqrt{}}\delta' = 0. \tag{B.39}$$

Letting $A = A_0 + \hat{A}$, $B = B_0 + \hat{B}$ with $A_0 = \partial_\sigma(v\mathcal{H}_R/R_v\sqrt{})$ and $B_0 = \partial_\sigma(v\mathcal{J}_-/R_v\sqrt{})$, we find the following constraint

$$\{\mathcal{H}_R, \tilde{\hat{B}}\} + \{\hat{A}, \tilde{\mathcal{J}}_-\} + \frac{\hat{B}}{\sqrt{}}\delta' = \frac{\lambda}{2R_v}\frac{\tilde{\mathcal{H}}_R}{\sqrt{}}\delta' - \frac{1}{R_v}\frac{\mathcal{J}_-}{\sqrt{}}\delta'. \tag{B.40}$$

The already known $\mathcal{H}_R/R_v$ contribution to $\hat{A}$ accounts for the last term on the right-hand side. However, we need a new term in $\hat{B}$ to account for the first term. It is clear that

$$\hat{B} = -\frac{\lambda\mathcal{H}_R}{2R_v} \tag{B.41}$$

does the job, and one can check that it is also consistent with the previous consistency conditions.

Finally, it is not hard to check that any term in $B$ proportional to $\partial_\sigma\mathcal{J}_-/\sqrt{}$ and in $A$ with $\partial_\sigma\mathcal{H}_R/\sqrt{}$ (with the same proportionality coefficient) automatically satisfies this equation. This sets $a = b$. We can also check the time dependence matches exactly.

To summarize, the solution that we have found for B that is consistent with all the Jacobi identities we have checked is

$$\boxed{B(\sigma) = \partial_\sigma\left[\left(\frac{v+t}{R_v} + a\right)\frac{\mathcal{J}_-}{\sqrt{}}\right] - \frac{\lambda\mathcal{H}_R}{2R_v}} \tag{B.42}$$

for the same arbitrary constant $a$ as in (B.27).

### B.3. Other commutators

The Poisson brackets of $\chi$ with all the remaining fields, such as $\mathcal{J}_+$ or $\mathcal{P}$, are determined by its commutators with $\mathcal{H}_R$ and $\mathcal{J}_-$ (i.e., the functions $A, B$) and its commutators with $\mathcal{H}_L = \mathcal{H}_R + \mathcal{P}$ and $K_U = \mathcal{J}_+ + \lambda\mathcal{H}_R/2$, which are local. We thus find

$$\{\mathcal{P}, \tilde{\chi}\} = -\left(\mathcal{H}_R + \frac{\mathcal{H}_R}{\sqrt{}}\right)\delta(\sigma-\tilde{\sigma}) + \partial_\sigma\frac{\mathcal{H}_R}{\sqrt{}}\Theta(\tilde{\sigma}-\sigma) + A(\sigma), \tag{B.43}$$

$$\{\chi, \tilde{\mathcal{J}}_+\} = \frac{\lambda \mathcal{H}_R}{2\sqrt{}} \delta(\sigma - \tilde{\sigma}) - \frac{\lambda}{2} \partial_{\tilde{\sigma}} \frac{\tilde{\mathcal{H}}_R}{\sqrt{}} \Theta(\sigma - \tilde{\sigma}) - \frac{\lambda}{2} A(\tilde{\sigma}). \tag{B.44}$$

Combining the latter with the $\{\chi, \tilde{\mathcal{J}}_-\}$ commutator and integrating, we find the $\{\chi, \tilde{\phi}\}$ commutator

$$\{\chi, \tilde{\phi}\} = -\frac{\tilde{\mathcal{J}}_- + \lambda \tilde{\mathcal{H}}_R/2}{\sqrt{}} \left( \Theta(\sigma - \tilde{\sigma}) + \frac{\tilde{v} + \tilde{t}}{R_v} + a \right) + C(\sigma). \tag{B.45}$$

The $\{\mathcal{H}_R, \tilde{\phi}\}$ commutator requires that $C'(\sigma) = 0$, so $C = c$, a constant, which we will set to zero.

## C. Flow equations for the currents and the charges

Let $\mathcal{D}_\lambda = \partial_\lambda + \{\mathcal{O}_{tot}, \cdot\}$, where for the purposes of computing commutators, the expression

$$\mathcal{O}_{tot} = \frac{w\chi_0}{R} - \int d\sigma \, \mathcal{H}_R \hat{\phi} \tag{C.1}$$

is significantly easier to use. Using the commutators with the zero modes, we find that at *classical* level, the currents satisfy the following flow equations

$$\mathcal{D}_\lambda K_U = \mathcal{D}_\lambda \mathcal{H}_L = 0. \tag{C.2}$$

The equation for $\mathcal{H}_R$ is

$$\mathcal{D}_\lambda \mathcal{H}_R = \frac{w}{R_v} \mathcal{H}_R + \partial_\sigma \left[ \frac{\mathcal{H}_R}{\sqrt{}} \left( w(1+a) - \frac{R\hat{\phi}}{R_v} \right) \right], \tag{C.3}$$

which implies the following flow equation for $\chi$

$$\mathcal{D}_\lambda \chi = \frac{\mathcal{H}_R}{\sqrt{}} \left( w(1+a) - \frac{R\hat{\phi}}{R_v} \right) + \frac{w}{R_v} \chi. \tag{C.4}$$

The flow equation for $\mathcal{J}_-$ is given by

$$\mathcal{D}_\lambda \mathcal{J}_- = -\frac{1}{2} \mathcal{H}_R - \frac{w\lambda}{2R_v} \mathcal{H}_R + \partial_\sigma \left[ \frac{\mathcal{J}_-}{\sqrt{}} \left( w(1+a) - \frac{R\hat{\phi}}{R_v} \right) \right], \tag{C.5}$$

which, together with (C.3), implies that

$$\mathcal{D}_\lambda (\mathcal{J}_- + \frac{\lambda}{2} \mathcal{H}_R) = \partial_\sigma \left[ \frac{\mathcal{J}_- + \lambda \mathcal{H}_R/2}{\sqrt{}} \left( w(1+a) - \frac{R\hat{\phi}}{R_v} \right) \right]. \tag{C.6}$$

Finally, the flow equation for $\phi$ is

$$\mathcal{D}_\lambda \phi(\sigma) = -\frac{\mathcal{J}_- + \lambda \mathcal{H}_R/2}{\sqrt{}} \left( w(1+a) - \frac{R\hat{\phi}}{R_v} \right). \tag{C.7}$$

We subsequently use these flow equations to compute the flow of the conserved charges. We trivially have $\mathcal{D}_\lambda Q_f = \mathcal{D}_\lambda P_\eta = 0$. As for the right-moving charges, we obtain

$$\mathcal{D}_\lambda \bar{Q}_{\tilde{f}} = \frac{w}{R_v} \bar{Q}_{\tilde{f}} - \frac{w}{R_v} \left( a + 1 + \frac{t}{R_v} \right) \bar{Q}_{\tilde{f}'}, \qquad \mathcal{D}_\lambda \bar{P}_{\tilde{\eta}}^{KM} = -\frac{w}{R_v} \left( a + 1 + \frac{t}{R_v} \right) \bar{P}_{\tilde{\eta}'}^{KM}. \tag{C.8}$$



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
