# Peer review of "Symmetries versus the spectrum of $ J\bar T$-deformed CFTs"

_SciPost Physics, doi:SciPost Phys. 10, 065 (2021)_

## Round 2 · Referee Report · Anonymous · 2021-2-11

Report

This work is a direct continuation of previous works of the author
on this topic, which deals with subtleties concerning symmetries
of solvable irrelevant deformed CFTs, and addresses new ideas
relevant for their resolution.

It is suitable for publication

---

## Round 2 · Referee Report · Anonymous · 2021-2-28

Strengths

see report

Weaknesses

see report

Report

This manuscript studies the apparent tension between the deformed energy level formula for a $J\bar{T}$ deformed CFT on the cylinder and the equally-spaced energies that follow from the symmetries. It is shown that the previously found infinite set of symmetry generators (which are field-dependent) does not act properly on the semiclassical phase space of the theory, but a new (infinite) set of symmetry generators does. These new generators preserve the algebra and have the correct charge and momentum quantization.

The manuscript is well-written, although sometimes a bit formal, and addresses a very important relevant issue in the $J\bar{T}$ literature. The proposed solution is valuable, not only for the $J\bar{T}$ deformation, but potentially also other deformations such as the $T\bar{T}$ deformation.

One confusion I had was with the rather formal expressions 3.40 and 3.49. Is it clear that these sums are convergent? Is the radius of convergence related to the complexification of the energy levels in 1.1?

Typo: It is Kac and not Ka\v{c}

Other than this minor confusion I recommend this manuscript for publication.

---

## Editorial Decision

published